# SRAS1.1 E3 ligase mediates DSK2A degradation to regulate autophagy and drought tolerance in *Arabidopsis*

Xiao-Hu Li [1,3], Meng Wang [1,3], Yi-Ran Xu[1,3], Qian-Huan Guo[1], Peng Liu [2], Chang-Ai Wu [1], Guo-Dong Yang [1], Jin-Guang Huang[1], Shi-Zhong Zhang [1✉], Cheng-Chao Zheng [1✉] & Kang Yan [1✉]

## Abstract

Drought stress significantly impacts plant growth and productivity, requiring complex adaptive responses to ensure survival. In eukaryotes, autophagy and the ubiquitin-proteasome system (UPS) are critical pathways for maintaining cellular homeostasis under stress. While their interaction is well-studied in animals, it remains poorly understood in plants, particularly under drought conditions. Here, we identify the E3 ubiquitin ligase SRAS1.1 as a key regulator of selective autophagy and drought tolerance in *Arabidopsis*, mediating its function through the ubiquitination and degradation of the autophagy receptor DSK2A. Loss of SRAS1.1 enhances drought tolerance by reducing water loss, increasing survival rates, and accelerating flowering. SRAS1.1 directly interacts with and ubiquitinates the autophagy receptor DSK2A, promoting its degradation via the 26S proteasome. Notably, under drought stress, SRAS1.1 relocates from the nucleus to the cytoplasm, associates with autophagosomes, and modulates autophagy-related gene expression and BES1 accumulation. These findings provide novel insights into UPS-autophagy crosstalk in plants and highlight SRAS1.1 as a promising target for genetic engineering to develop drought-resilient crops and to advance sustainable agriculture.

Keywords Drought Stress; E3 Ubiquitin Ligase; Autophagy; DSK2A; Stress Tolerance
Subject Categories Autophagy & Cell Death; Plant Biology; Post-translational Modifications & Proteolysis

## Introduction

Drought represents a significant environmental constraint on crop production, severely impacting plant growth and development (Gao et al, 2022; Gupta et al, 2020; Zhang et al, 2022). To cope with deleterious drought stress, plants have evolved a variety of avoidance and acclimation mechanisms, such as reducing water loss, closing stomata, accelerating flowering, and triggering the expression of stress-related genes (Gupta et al, 2020; Waadt et al, 2022). Various regulatory factors contribute to these processes, including transcription factors, phytohormones, protein kinases, and ubiquitin ligases (Chen et al, 2021a; Chen et al, 2021b; Dong et al, 2020). However, the molecular mechanisms underlying plant drought adaptation remain incompletely understood, which limits progress in breeding early-maturing and drought-tolerant crop varieties (Chong et al, 2022; Verslues et al, 2023; Waadt et al, 2022).

Autophagy, an evolutionarily conserved degradative mechanism in eukaryotic cells, occurs constitutively at a basal level for maintenance of cellular homeostasis (Bao et al, 2020; Liu et al, 2023a; Sun et al, 2023). Autophagy is triggered by various stresses, including pathogen infections, nutrient starvation, and environmental stress, including drought stress, facilitating the degradation of intracellular components into metabolites to sustain cell survival (Huang et al, 2023; Tang and Bassham, 2021; Zhou et al, 2023). Recent studies have uncovered a close relationship between drought stress and autophagy. A plant-specific gene *Constitutively stressed 1* (*COST1*) negatively regulates plant drought resistance by influencing the autophagy pathway (Bao et al, 2020). Autophagy receptor NEXT TO BRCA1 GENE 1 (ZmNBR1) promotes the autophagic degradation of BRASSINOSTEROID INSENSITIVE 1A (ZmBRI1a) and enhances drought tolerance in maize (Xiang et al, 2024).

In eukaryotes, the ubiquitin-proteasome system (UPS)-mediated degradation pathway represents another regulatory mechanism involved in modulating various cellular processes, including hormone signaling, DNA repair, and biotic and abiotic stress responses (Kaur et al, 2013; Martínez et al, 2024; Yu et al, 2024). E3 ubiquitin ligases, as important components of the UPS, catalyze the binding of ubiquitin to their target proteins via sequential actions from E2 ubiquitin-conjugating enzymes, thereby ensuring substrate specificity (Eckardt et al, 2024; Su et al, 2024; Tang et al, 2023). Under drought stress conditions, E3 ligases specifically recognize and ubiquitinate target proteins, thereby precisely regulating stress signaling pathways (Chen et al, 2021a; Chen et al, 2021b). E3 ligase Grain width and weight2 (TaGW2) ubiquitinates and degrades the

[1]State Key Laboratory of Wheat Improvement, College of Life Sciences, Shandong Agricultural University, 271018 Taian, Shandong, P. R. China. [2]Donald Danforth Plant Science Center, St. Louis, MO 63132, USA. [3]These authors contributed equally: Xiao-Hu Li, Meng Wang, Yi-Ran Xu. ✉E-mail: shizhong@sdau.edu.cn; cczheng@sdau.edu.cn; kangyan@sdau.edu.cn

*Arabidopsis* response regulator 12 (TaARR12), which reduces water loss and enhances wheat drought resistance (Li et al, 2024; Su et al, 2024). Additionally, DREB2A-INTERACTING PROTEIN1 (DRIP1), a RING domain–containing protein, interacts with and degrades DREB2A, thereby negatively regulating drought stress responses in plants (Qin et al, 2008; Zhang et al, 2017).

Interestingly, despite their distinct degradation mechanisms, the UPS and autophagy share molecular determinants and substrates. Emerging evidence highlights a dynamic cross-talk between the two pathways (Chen et al, 2019; Nag et al, 2023). In animals, the E3 ubiquitin ligase neuronal precursor cell-expressed, developmentally downregulated 4 (NEDD4) ubiquitinates and degrades a key regulator of autophagy Beclin 1, thereby suppressing cellular autophagy negatively regulates virus replication (Xu et al, 2017). The Gigaxonin E3 ligase governs the turnover of autophagy-related protein 16-1 (ATG16L1) to control autophagosome production and maintain cellular homeostasis (Scrivo et al, 2019). Although extensively studied in animals, the interaction between the UPS and autophagy remains poorly understood in plants, particularly under drought stress.

The DOMINANT SUPPRESSOR OF KAR 2A (DSK2A), which functions as both a ubiquitin receptor and an autophagy receptor, forms a complex with the E3 ubiquitin ligase SEVEN IN ABSENTIA OF ARABIDOPSIS 2 (SINAT2), leading to the degradation of BRI1-EMS-SUPPRESSOR1 (BES1). Concurrently, DSK2A recruits ATG8e to interact with BES1, thereby facilitating its autophagic degradation under drought stress (Nolan et al, 2017). In a previous study, we investigated the function of the RING-type E3 ligase Salt-responsive alternatively spliced gene 1 (SRAS1), which produces two transcript isoforms with opposing responses and distinct functions under salt stress (Zhou et al, 2021). Building on this foundation, here we demonstrated that drought-responsive SRAS1.1 directly targets DSK2A and promotes its degradation, thereby regulating the BES1 accumulation to control autophagy and modulate drought tolerance. Collectively, our study provides evidence that an E3 ligase can degrade an autophagy receptor, highlighting the integration of autophagy and the UPS in plant drought adaptation.

## Results

### E3 ubiquitin ligase SRAS1.1 negatively regulates plant drought tolerance

In our previous study, we investigated the role of *SRAS1* splicing variants. Transcriptome analyses revealed that the expression levels of many drought-response genes were altered in *SRAS1.1* over-expressing lines, including *ABI5*, *DREB1C*, and *WRKY33* (Zhou et al, 2021). To evaluate the contribution of SRAS1.1 to drought response, we performed quantitative RT-PCR (qRT-PCR) analysis. The transcript levels of *SRAS1.1* gradually decreased under drought stress conditions (Fig. 1A). Furthermore, GUS staining and GUS activity in *proSRAS1.1:GUS* transgenic seedlings significantly decreased after drought treatment, indicating that drought stress strongly suppresses the expression of *SRAS1.1* (Fig. 1B,C).

To investigate the role of *SRAS1.1* in response to drought stress, we analyzed the phenotypes of *SRAS1.1* overexpressing lines (*SRAS1.1-14*, *SRAS1.1-26*) and *sras1.1* mutants using physiological assays. A primary root growth assay showed that *SRAS1.1* overexpressing lines (*SRAS1.1-OE*) significantly reduced primary root length while increasing the lateral root density following drought treatment (Figs. 1D–F and EV1A–C). Germination rate assays indicated that *SRAS1.1-OE* exhibited significantly higher sensitivity to drought stress compared to both wild-type and *sras1.1* mutants (Fig. EV1D–I). Building on these findings, we further assessed the survival under prolonged drought conditions. After 16 days of water deprivation in a soil drought experiment, the survival rates of two independent *SRAS1.1-OE* lines dropped to 18.3% and 29.7%, respectively, with more pronounced leaf wilting, whereas *sras1.1* mutants showed a significantly higher survival rate of 79.4% (Fig. 1G,H). Complementation with *proSRAS1.1:SRAS1.1* restored drought-sensitive phenotypes, as evidenced by decreased germination rate, reduced root length, and lower survival compared to *sras1.1* mutants (Fig. EV1; Appendix Fig. S1). Water loss analysis in detached leaves indeed confirmed that the rate of water loss in *sras1.1* was slower than that in the wild-type, but *SRAS1.1-OE* lines lost more water than the wild-type in soil and in the detached leaf water loss assay (Fig. 1I). In the following stomatal closure assay under drought conditions, the stomatal aperture of *sras1.1* mutants was significantly smaller than that of wild-type plants (Fig. 1J,K). Meanwhile, the expression level of the *S-type anion channel* (*SLAC1*) was significantly increased in *sras1.1* (Fig. 1L), and the expression of drought resistance genes was reduced in *SRAS1.1-OE* plants (Fig. EV1J–M). These findings indicate that *SRAS1.1* negatively regulates drought resistance.

Accelerated reproduction and flowering are crucial strategies for plants to escape drought stress (Gupta et al, 2020; Xu et al, 2022). Under normal conditions, the flowering time of *SRAS1.1-OE* plants was 4.26 days earlier than that of wild-type plants. Under drought stress, flowering was significantly accelerated in *SRAS1.1-OE* plants (Fig. EV2A–C). Scanning electron microscopy (SEM) analysis of *SRAS1.1-OE* plants showed that some pollen grains were broken, collapsed, or deformed (Fig. EV2D). Pollen germination rates in *SRAS1.1-OE* plants were significantly lower than in wild-type plants (Fig. EV2E,F). Furthermore, expression analysis of flowering-related genes, including FLOWERING LOCUS T (*FT*), CON-STANS (*CO*), SUPPRESSOR OF OVEREXPRESSION OF CO 1 (*SOC1*), and LEAFY (*LFY*), revealed transcriptional changes consistent with the early flowering phenotype (Fig. EV2G–J). These findings suggest that while *SRAS1.1* promotes flowering under drought stress, its effects on pollen viability may impair reproductive success.

### SRAS1.1 modulates drought-responsive gene expression

To uncover the molecular mechanism of *SRAS1.1* under drought stress, we conducted RNA-seq analysis on 10-day-old wild-type, *sras1.1* mutants, and *SRAS1.1-14* seedlings treated with 250 mM mannitol. Three biological replicates were sequenced for each growth condition. The average read length for sequences mapped to the *Arabidopsis* reference genome (TAIR10) was approximately 101 bp. Each library consisted of 23-24 million reads, of which ~90% mapped to unique loci. A total of 119 differentially expressed genes (DEGs) were identified in *sras1.1* mutants compared to the wild-type, comprising 50 upregulated and 69 downregulated genes (Fig. 2A; Dataset EV1). Similarly, 518 DEGs were detected between *SRAS1.1-14* and wild-type plants, comprising 323 upregulated and

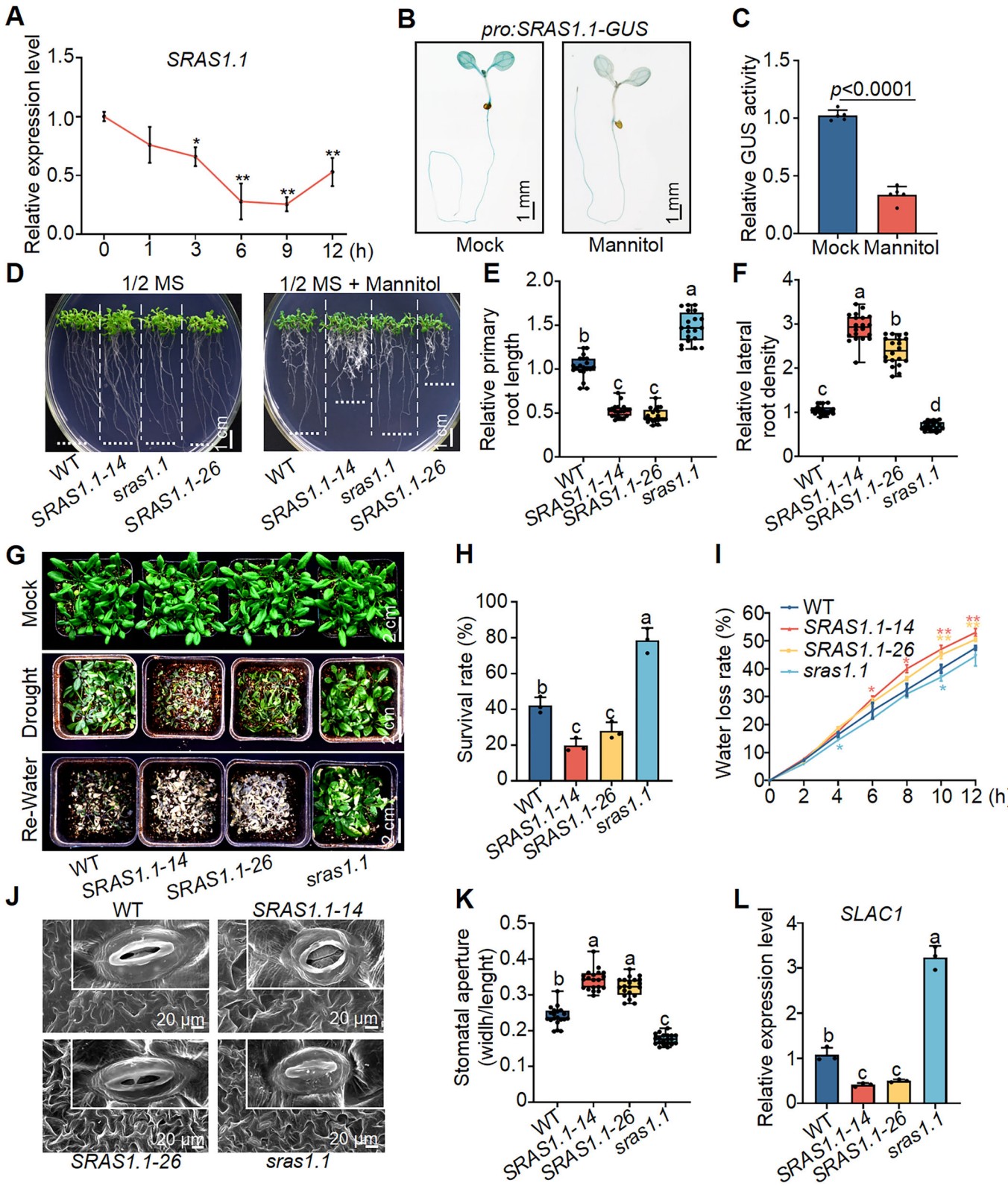

**Figure 1. SRAS1.1 negatively regulates drought tolerance in *Arabidopsis*.**

(A) Quantitative measurement of the *SRAS1.1* transcript level in 2-week-old wild-type (Col-0) plants after drought stress for 0, 1, 3, 6, 9, or 12 h. *UBQ10* was used as an internal control. Values shown are means ± SD ($n = 3$ biological replicates). *P* values = 0.0725 (0 h vs 1 h), 0.0131 (0 h vs 3 h), 0.0022 (0 h vs 6 h), 0.0017 (0 h vs 9 h), 0.0042 (0 h vs 12 h). (B, C) GUS staining (B) and quantitative activity analysis of *proSRAS1.1:GUS* seedlings (C) under mock and mannitol treatment. Scale bars = 1 mm. Values shown are means ± SD ($n = 5$ biological replicates). Significance was determined using Student's *t* test. *P* values < 0.0001. (D–F) Representative seedlings of wild-type, *SRAS1.1*-overexpressing lines (*SRAS1.1-OE*), and *sras1.1* mutants after 2 weeks of growth in 1/2 MS medium with or without 250 mM mannitol. Scale bars = 1 cm. Primary root length (E) and lateral root density (F) of seedlings shown in (D). Both graphs are presented as the percentage relative to growth on control 1/2 MS medium, and it was designated as 1 in wild-type. Values shown are means ± SD ($n = 3$ biological replicates), with 20 plants analyzed per replicate. (E) *P* values < 0.0001 (wild-type vs *SRAS1.1-14*), <0.0001 (wild-type vs *SRAS1.1-26*), <0.0001 (wild-type vs *sras1.1*), 0.4869 (*SRAS1.1-14* vs *SRAS1.1-26*), <0.0001 (*SRAS1.1-14* vs *sras1.1*), <0.0001 (*SRAS1.1-26* vs *sras1.1*). (The following is the same order). (F) *P* values < 0.0001, <0.0001, <0.0001, <0.0001, <0.0001, <0.0001. (G) Morphology of seedlings before and after drought stress treatment. 2-week-old wild-type, *SRAS1.1-OE*, and *sras1.1* mutant plants were subjected to drought stress for 16 days and then rewatered for 5 days. Scale bars = 2 cm. (H) Survival rate after drought treatment. Values shown are means ± SD ($n = 3$ biological replicates). *P* values = 0.0032 (wild-type vs *SRAS1.1-14*), 0.038 (wild-type vs *SRAS1.1-26*), 0.0001 (wild-type vs *sras1.1*), 0.2931 (*SRAS1.1-14* vs *SRAS1.1-26*), <0.0001 (*SRAS1.1-14* vs *sras1.1*), <0.0001 (*SRAS1.1-26* vs *sras1.1*). (I) Water loss rate in detached leaves of 3-week-old wild-type, *SRAS1.1-OE*, and *sras1.1* mutant plants. Values shown are means ± SD ($n = 3$ biological replicates), with 10 detached leaves analyzed per replicate. Significance was determined using Student's *t* test. T = 2 h: *P* values = 0.0929 (wild-type vs *SRAS1.1-14*), 0.3045 (wild-type vs *SRAS1.1-26*), 0.1631 (wild-type vs *sras1.1*). (The following is the same order). T = 4 h: *P* values = 0.0651, 0.1448, 0.0414. T = 6 h: *P* values = 0.0361, 0.1401, 0.2472. T = 8 h: *P* values = 0.0112, 0.1813, 0.1428. T = 10 h: *P* values = 0.0012, 0.0054, 0.0149. T = 12 h: *P* values = 0.0219, 0.0201, 0.0739. (J) Representative images of stomata in wild-type, *SRAS1.1-OE*, and *sras1.1* mutants after drought stress. Scale bars = 20 μm. (K) Quantification of stomatal closure in different genotypes shown in (J). Values shown are means ± SD ($n = 3$ biological replicates), with 20 plants analyzed per replicate. *P* values < 0.0001 (wild-type vs *SRAS1.1-14*), <0.0001 (wild-type vs *SRAS1.1-26*), <0.0001 (wild-type vs *sras1.1*), 0.078 (*SRAS1.1-14* vs *SRAS1.1-26*), <0.0001 (*SRAS1.1-14* vs *sras1.1*), <0.0001 (*SRAS1.1-26* vs *sras1.1*). (L) Quantitative measurement of the expression levels of *SLAC1*. The data were normalized against *UBQ10* expression. Values shown are means ± SD ($n = 3$ biological replicates). *P* values = 0.0031 (wild-type vs *SRAS1.1-14*), 0.0073 (wild-type vs *SRAS1.1-26*), <0.0001 (wild-type vs *sras1.1*), 0.8888 (*SRAS1.1-14* vs *SRAS1.1-26*), <0.0001 (*SRAS1.1-14* vs *sras1.1*), <0.0001 (*SRAS1.1-26* vs *sras1.1*). Data information: For (A, C, I) values shown are means ± SD. Asterisks represent significant differences determined by Student's *t* test (*$P < 0.05$; **$P < 0.01$). For (E, F, H, K, L) different lowercase letters represent significant differences, as determined by one-way ANOVA in combination with Tukey's multiple comparisons test ($P < 0.05$). Data in (E, F, K) are plotted with box–whisker plots: the whiskers represent maximum and minimum values, and boxes represent the upper quartile, median, and lower quartile, dots represent data points. Source data are available online for this figure.

195 downregulated genes (Fig. 2B,C; Dataset EV2). Gene Ontology (http://www.geneontology.org/) and Kyoto Encyclopedia of Genes and Genomes (http://www.genome.jp/kegg) enrichment analyses revealed that these DEGs were primarily involved in stress and hormone responses, including brassinosteroid signaling, chemical stimuli, and hormone signal transduction (Fig. 2D,E). To further confirm these findings, we validated the expression of selected DEGs by qRT-PCR. Consistent with the RNA-seq data, drought stress-related positive regulators, such as *RD20* and *RD22*, were upregulated in the *sras1.1* mutants (Fig. 2F,G). Conversely, the expression of negative stress regulators, such as *WRKY54* and *WRKY70*, was downregulated in the *sras1.1* mutants (Fig. 2H,I). These results are consistent with the role of *SRAS1.1* as a negative regulator of drought stress.

## SRAS1.1 interacts with DSK2A

To investigate how SRAS1.1 directly regulates plant drought responses, we performed immunoprecipitation coupled with mass spectrometry (IP-MS) to identify SRAS1.1-interacting proteins under drought conditions. Among the 216 candidate interactors, we focused on the ubiquitin receptor DSK2A and its homolog DSK2B (Appendix Fig. S2; Dataset EV3). DSK2A and DSK2B are known to be involved in plant drought responses and brassinolide signal transduction, which aligns with SRAS1.1 functions (Fig. 2D,E).

To determine whether SRAS1.1 physically interacts with DSK2A and DSK2B, we conducted a series of protein-protein interaction assays. Yeast two-hybrid (Y2H) and luciferase complementation imaging (LCI) assays confirmed the interaction between SRAS1.1 and DSK2A/DSK2B in vitro (Fig. 3A–C; Appendix Fig. S3). To further validate these interactions, we conducted Co-immunoprecipitation (Co-IP) assays by co-expressing SRAS1.1-GFP with either DSK2A-HA or DSK2B-HA in *N. benthamiana*

leaves. These results confirmed the physical interactions between SRAS1.1 and both DSK2A and DSK2B (Fig. 3D). Bimolecular fluorescence complementation (BiFC) assays revealed that SRAS1.1-cYFP and nYFP-DSK2A reconstituted fluorescence in both the nucleus and cytoplasm, whereas SRAS1.1-cYFP and nYFP-DSK2B showed fluorescence primarily in the cytoplasm (Fig. 3E). Taken together, these results demonstrate that SRAS1.1 interacts with DSK2A and DSK2B both in vitro and in vivo.

## SRAS1.1 mediates DSK2A degradation through K⁶⁸-dependent ubiquitination

Our previous study revealed that *SRAS1.1* encodes an E3 ubiquitin ligase responsible for mediating protein degradation (Zhou et al, 2021). To investigate whether SRAS1.1 influences the stability of DSK2A and DSK2B, we assessed the protein levels of DSK2A and DSK2B. The results indicated that the protein level of DSK2A in the *SRAS1.1* overexpressing lines was lower compared to the wild-type, while it was significantly higher in the *sras1.1* mutants than in the wild-type (Fig. 4A). In contrast, DSK2B protein levels remained unchanged, highlighting the specificity of SRAS1.1 for DSK2A (Fig. 4A). Next, we tested whether SRAS1.1 ubiquitinates DSK2A and DSK2B through an in vitro ubiquitination assay using purified recombinant DSK2A-His and DSK2B-His along with GST-SRAS1.1. In the presence of ubiquitin, E1 and E2, recombinant GST-SRAS1.1 ubiquitinated DSK2A-His, whereas DSK2B-His was not ubiquitinated (Fig. 4B). To further confirm the in vivo function of SRAS1.1, we transiently expressed *DSK2A-GFP* or *DSK2B-GFP*, with or without ubiquitin-MYC, in wild-type and *sras1.1* plants. The ubiquitination level of *DSK2A-GFP* was markedly reduced in the *sras1.1* mutants compared to wild-type, while *DSK2B-GFP* showed no significant difference, indicating that SRAS1.1 specifically mediates DSK2A ubiquitination in planta (Fig. EV3A).

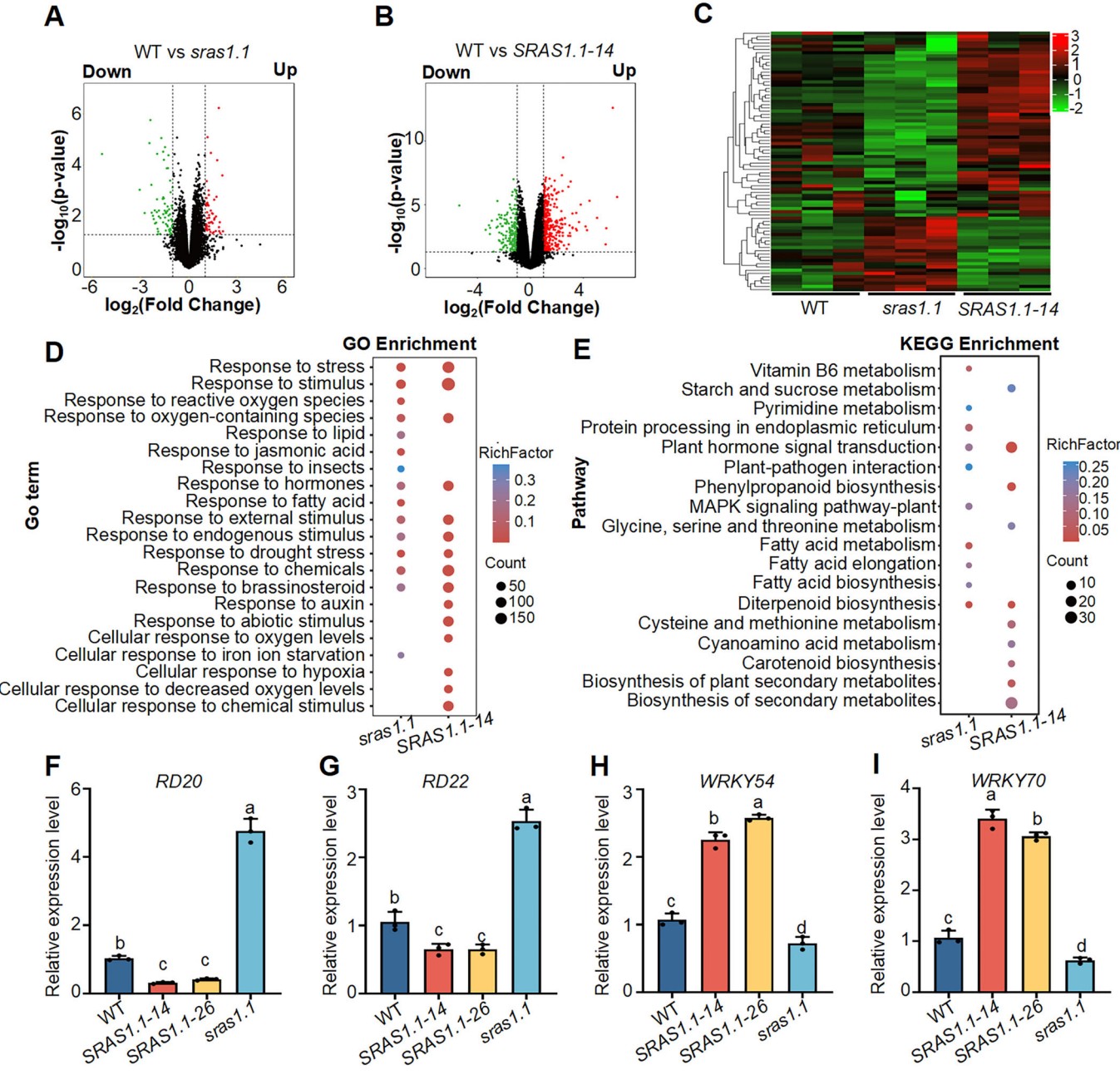

**Figure 2. SRAS1.1 mutation enhances drought-responsive gene expression.**

(A, B) Volcano plots illustrating the distribution of differentially expressed genes (DEGs) in *sras1.1* mutants (A) and *SRAS1.1-14* (B) compared to wild-type, with upregulated (red) and downregulated (green) genes. Differential expression was analyzed using DESeq2. Genes with an adjusted P value (FDR) < 0.01 and |log2(fold change)| > 1 were defined as significantly differentially expressed.The volcano plot shows $-\log_{10}$(P value) versus $\log_2$(fold change). Data are based on three independent biological replicates (*n* = 3). (C) Heatmap showing hierarchical clustering of DEGs in wild-type, *sras1.1* mutants and *SRAS1.1-14* plants. Colors represent normalized gene expression values. (D) Gene Ontology (GO) enrichment analysis of DEGs in *sras1.1* mutants and *SRAS1.1-14* compared to wild-type after 6 h of 250 mM mannitol treatment. Circle size represents the number of enriched genes, and circle color denotes enrichment significance. (E) Kyoto Encyclopedia of Genes and Genomes (KEGG) enrichment analysis of DEGs in *sras1.1* mutants and *SRAS1.1-14* compared to wild-type after 6 h of 250 mM mannitol treatment. Circle size indicates the number of enriched genes, and color represents enrichment significance. (F–I) Quantitative measurement of the expression levels of drought-responsive genes *RD20* (F), *RD22* (G), *WRKY54* (H), *WRKY70* (I). Expression levels were normalized to *UBQ10*. Values shown are means ± SD (*n* = 3 biological replicates). Statistical analysis was performed using one-way ANOVA followed by Tukey's multiple comparisons test. Different lowercase letters indicate significant differences (P < 0.05). (F) P values = 0.0052 (wild-type vs *SRAS1.1-14*), 0.0128 (wild-type vs *SRAS1.1-26*), <0.0001 (wild-type vs *sras1.1*), 0.8928 (*SRAS1.1-14* vs *SRAS1.1-26*), <0.0001 (*SRAS1.1-14* vs *sras1.1*), <0.0001 (*SRAS1.1-26* vs *sras1.1*). (The following is the same order). (G) P values = 0.017, 0.017, <0.0001, >0.9999, <0.0001, <0.0001. (H) P values < 0.0001, <0.0001, 0.0066, 0.0099, <0.0001, <0.0001. (I) P values < 0.0001, <0.0001, 0.009, 0.0362, <0.0001, <0.0001. Source data are available online for this figure.

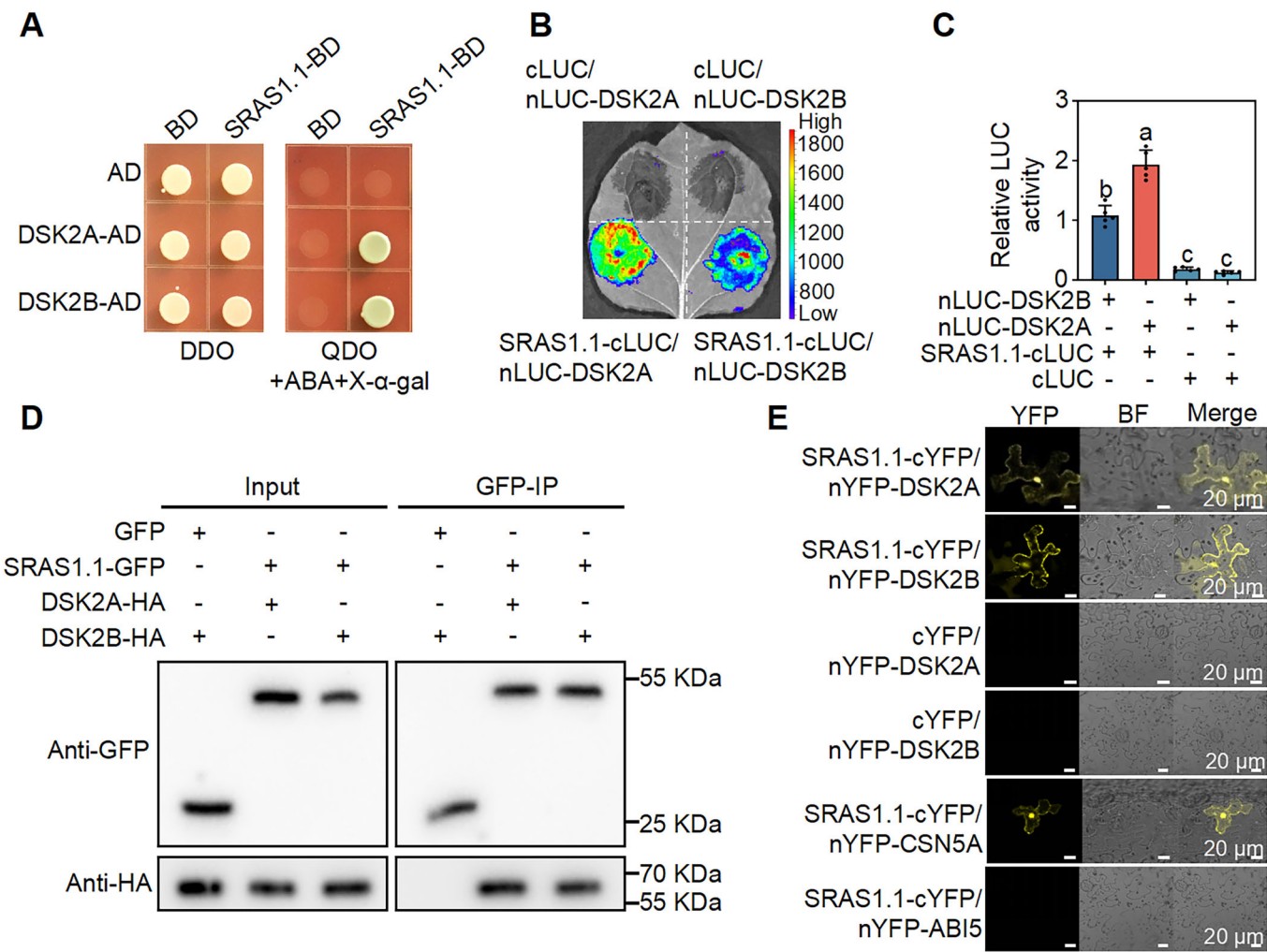

**Figure 3. SRAS1.1 selectively interacts with DSK2A.**

(A) Yeast two-hybrid (Y2H) analysis confirmed the interaction between SRAS1.1 and both DSK2A and DSK2B. Interaction between empty AD and BD vectors was used as a negative control. (B, C) Luciferase complementation imaging assay (LCI) (B) and LUC activity measurement (C) showed that SRAS1.1 interacts with DSK2A and DSK2B. Interaction between empty cLUC vector was used as a negative control. Values shown are means ± SD ($n = 5$ biological replicates). Statistical analysis was performed using one-way ANOVA followed by Tukey's multiple comparisons test. Different lowercase letters indicate significant differences ($P < 0.05$). $P$ values < 0.0001 (nLUC-DSK2B + SRAS1.1-cLUC vs nLUC-DSK2A + SRAS1.1-cLUC), <0.0001 (nLUC-DSK2B + SRAS1.1-cLUC vs nLUC-DSK2B + cLUC), <0.0001 (nLUC-DSK2B + SRAS1.1-cLUC vs nLUC-DSK2A + cLUC), <0.0001 (nLUC-DSK2A + SRAS1.1-cLUC vs nLUC-DSK2B + cLUC), <0.0001 (nLUC-DSK2A + SRAS1.1-cLUC vs nLUC-DSK2A + cLUC), 0.9521 (nLUC-DSK2B + cLUC vs nLUC-DSK2A + cLUC). (D) Co-immunoprecipitation (Co-IP) assay showing the interaction of SRAS1.1 with DSK2A and DSK2B in vivo. Immunoprecipitated proteins were analyzed by immunoblotting using anti-GFP and anti-HA antibodies. (E) Bimolecular fluorescence complementation (BiFC) assay visualized interactions between SRAS1.1 and DSK2A/DSK2B. YFP signals are detected in both the nucleus and cytoplasm for SRAS1.1-DSK2A, and predominantly in the cytoplasm for SRAS1.1-DSK2B. The interaction between SRAS1.1 and empty nYFP-ABI5 was used as a negative control, and nYFP-CSN5A as a positive control. Scale bars = 20 μm. Source data are available online for this figure.

Additionally, robust ubiquitination of DSK2A-His was observed, confirming that DSK2A is a direct target of SRAS1.1. To determine whether SRAS1.1 mediates the degradation of DSK2A and DSK2B, we performed cell-free degradation assays to test their stability. Immunoblot analysis showed that DSK2A-His was degraded after an extended time of ATP application in both the wild-type, *SRAS1.1-14* and *sras1.1* mutants. However, this degradation occurred at a significantly slower rate in the *sras1.1* mutants (Fig. 4C,E). Conversely, the degradation of DSK2B-His remained unchanged between wild-type and *SRAS1.1-14* plants (Fig. 4D,F). These results suggest that SRAS1.1 selectively promotes the ubiquitination and degradation of DSK2A.

To pinpoint the ubiquitination site in DSK2A, we identified four lysine (K) residues as potential targets (Fig. EV3B). Using the predictor of protein ubiquitination sites (https://www.psb.ugent.be), $K^{68}$ was the most probable ubiquitination site in DSK2A (Fig. EV3C). To test this hypothesis, we generated two variants of DSK2A, mimicking the non-ubiquitinated state by mutating these residues to arginine (R) in $DSK2A^{K29R}$ and $DSK2A^{K68R}$. To evaluate whether SRAS1.1 promotes ubiquitination at these sites, we conducted a cell-free degradation assay using both wild-type and mutant DSK2A-His proteins. Compared to that of the control, the wild-type version of DSK2A was ubiquitinated by

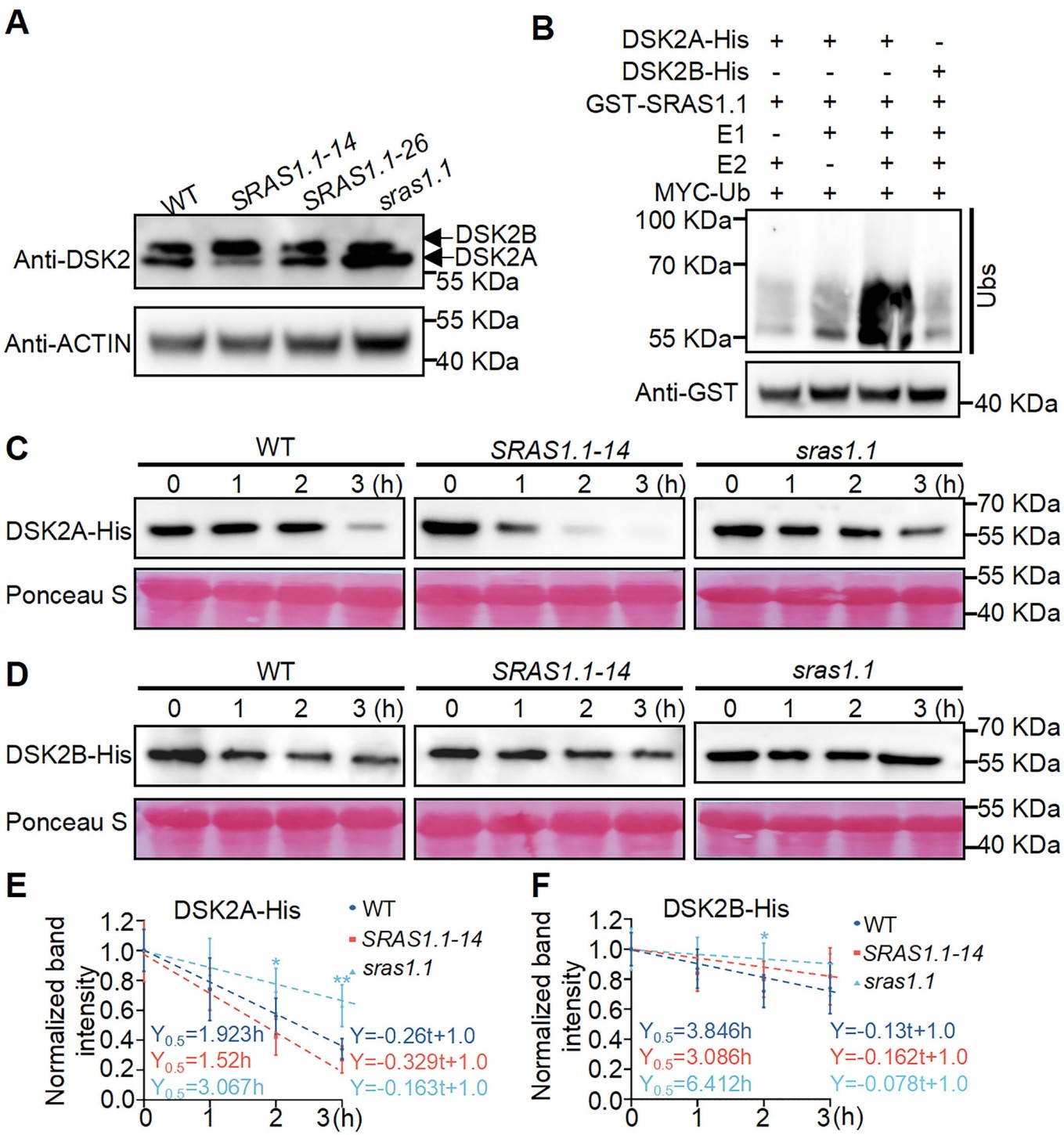

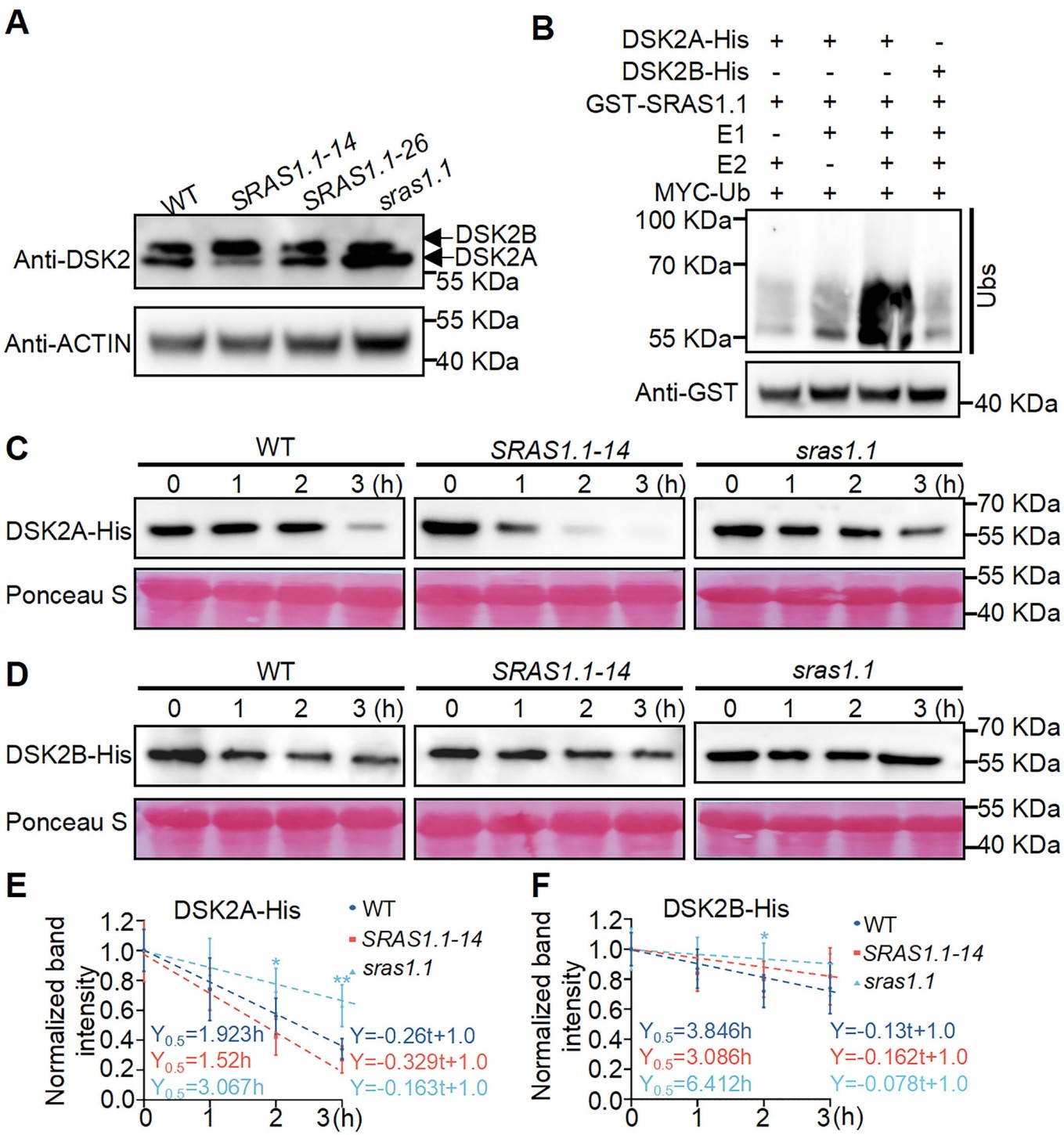

SRAS1.1 in both the wild-type, *SRAS1.1-14* and *sras1.1* mutants plants. However, when the lysine residue $K^{68}$ was mutated to arginine ($K^{68}R$), the degradation of $DSK2A^{K68R}$ was significantly reduced compared to DSK2A or $DSK2A^{K29R}$ in wild-type, *SRAS1.1-14*, and *sras1.1* mutants (Fig. EV3D–G). These findings indicate that SRAS1.1 specifically ubiquitinates DSK2A at $K^{68}$ to mediate its degradation.

## Drought stress inhibits SRAS1.1 mediated degradation of DSK2A

While DSK2A is known to act as a selective regulator balancing plant growth and survival under drought stress (Nolan et al, 2017), the molecular mechanisms by which drought stress modulates DSK2A activity, particularly its degradation, remain unclear. Addressing this

◀ **Figure 4. SRAS1.1 specifically ubiquitinates and promotes the degradation of DSK2A.**

(A) Western blot analysis of DSK2A and DSK2B protein abundance in wild-type, *SRAS1.1-OE*, and *sras1.1* mutant seedlings. To demonstrate the specificity of the Anti-DSK2 antibody, ACTIN was used as a loading control. (B) In vitro ubiquitination assay showed that DSK2A-His protein can be ubiquitinated by GST-SRAS1.1. Recombinant proteins DSK2A-His, DSK2B-His, and GST-SRAS1.1 purified from *E. coli* were assayed. Reaction products were analyzed by immunoblotting with Anti-MYC and Anti-GST. (C, D) Degradation rates of DSK2A-His (C) and DSK2B-His (D) were assessed in cell-free degradation assay using wild-type, *SRAS1.1-14* and *sras1.1* protein extracts. Recombinant purified DSK2A-His and DSK2B-His were added to the protein extracts and incubated for 1, 2, or 3 h. Protein abundance was determined with anti-His antibody. Similar results were obtained in 3 independent experiments. (E, F) Linear regressions of the quantified band intensities from (C, D) by ImageJ representing the degradation rates of DSK2A (E) and DSK2B (F) in wild-type, *SRAS1.1-14* and *sras1.1* protein extracts. $Y_{0.5}$ represents the time point at which half of the protein was degraded. Similar results were obtained from three independent experiments. Each figure panel displays a representative image obtained from a gel-based assay. All plant materials were compared with wild-type separately. Values shown are means ± SD ($n = 3$ biological replicates). All comparisons were made to wild-type plants. Statistical significance was determined using Student's *t* test (E) T = 1 h: *P* values = 0.2188 (wild-type vs *SRAS1.1-14*), 0.3464 (wild-type vs *sras1.1*). (The following is the same order). T = 2 h: *P* values = 0.0625, 0.0453. T = 3 h: *P* values = 0.0789, 0.0041. (F) T = 1 h: *P* values = 0.953, 0.1862. T = 2 h: *P* values = 0.3096, 0.0148. T = 3 h: *P* values = 0.6692, 0.1699. Data information: Values are presented as means ± SD ($n = 3$ biological replicates). All comparisons were made to wild-type plants. Statistical significance was determined using Student's *t* test (*$P < 0.05$; **$P < 0.01$). Source data are available online for this figure.

question is critical to understanding the upstream regulatory pathways involved in plant drought responses. To explore the genetic interaction between SRAS1.1 and DSK2A, we generated *sras1.1 dsk2a* and *sras1.1 dsk2b* double mutants through genetic crossing (Appendix Fig. S4A,B). Primary root growth assays showed that under drought treatment, the *dsk2a* mutants exhibited significantly shorter primary roots and increased lateral root density. The *sras1.1 dsk2a* double mutants displayed a similar drought sensitivity to the *dsk2a* single mutant (Fig. 5A–C). Germination rate assays revealed that *sras1.1 dsk2a* exhibited a drought-sensitive phenotype similar to the *dsk2a* single mutant, with significantly reduced germination under drought stress (Appendix Fig. S4C–F). Soil drought experiment demonstrated that the survival rate and water loss rate of *sras1.1 dsk2a* were similar to those of the *dsk2a* single mutants but distinct from *sras1.1* mutants (Fig. 5D–F). In contrast, the *dsk2b* single mutants and the *sras1.1 dsk2b* double mutants did not exhibit drought sensitivity comparable to that of the *dsk2a* mutants (Fig. 5D–F). Stomatal aperture analysis revealed that under drought stress, both the *sras1.1 dsk2a* double mutants and the *dsk2a* single mutants had significantly larger stomatal apertures compared to the wild-type. In contrast, the *dsk2b* single mutants and the *sras1.1 dsk2b* double mutants exhibited smaller stomatal apertures after drought treatment (Appendix Fig. S4G–I). Flowering time analysis under drought stress further revealed that the *sras1.1 dsk2a* double mutants flowered at the same time as the *dsk2a* mutants and significantly earlier than the wild-type (Appendix Fig. S5). This accelerated flowering may represent an adaptive strategy to mitigate drought impact. These results demonstrate that DSK2A is epistatic to SRAS1.1 in regulating drought tolerance.

Since SRAS1.1 functions upstream of DSK2A in regulating drought tolerance, we aimed to investigate whether drought stress influences the degradation of DSK2A by SRAS1.1. To test this hypothesis, we generated transgenic plants *35S:DSK2A-GFP* in the wild-type (*35S:DSK2A-GFP*) and *sras1.1* (*35S:DSK2A-GFP/sras1.1*) backgrounds and treated them with protein synthesis inhibitor cycloheximide (CHX), to further examine the effect of SRAS1.1 on DSK2A degradation. The degradation of DSK2A was reduced in the *sras1.1* mutants compared to the wild-type and was largely suppressed by the 26S proteasome inhibitor MG132 (Fig. 5G–I). Together, these results demonstrate that SRAS1.1 targets DSK2A for degradation via the 26S proteasome pathway. Despite identifying SRAS1.1 as a key regulator of DSK2A degradation under standard conditions, it remains unclear how drought stress influences this process. To address this, we utilized *35S:DSK2A-GFP* transgenic plants to assess whether SRAS1.1 mediates the

degradation of DSK2A proteins in response to drought stress. Under control conditions, the half-life of DSK2A proteins was measured at 4.13 h, while following treatment with 250 mM mannitol, the half-life increased to 6.08 h (Fig. 5G–I). Our results reveal that drought stress inhibits SRAS1.1-mediated degradation of DSK2A, possibly by affecting its activity.

## SRAS1.1 relocates to the cytoplasm and facilitates autophagosome formation under drought stress

Drought stress triggers significant cellular and molecular changes, including autophagosome formation, which is essential for maintaining cellular homeostasis Given that SRAS1.1 has been identified as a key regulator of drought tolerance, it remains unclear whether and how drought stress affects SRAS1.1 localization and its potential involvement in autophagy. To address this, we analyzed *35S:SRAS1.1-GFP* transgenic plants under drought conditions. In the mock-treated sample (0 mM mannitol), *35S:SRAS1.1-GFP* predominantly localized in the nucleus and cytoplasm. Upon treatment with 250 mM mannitol, SRAS1.1 exhibited increased cytoplasmic accumulation, along with the formation of distinct punctate structures in the cytoplasm (Fig. 6A). Consistently, in *35S:SRAS1.1-GFP* stable transgenic plants, drought stress induced the formation of punctate cytoplasmic structures, further supporting the dynamic relocalization of SRAS1.1 (Fig. EV4A,B). Given that DSK2A functions as an autophagy receptor under drought stress, we hypothesized that these punctate structures might represent autophagosomes. To confirm this hypothesis, we treated the samples with an the autophagy inhibitor concanamycin A (ConA). ConA treatment caused the disappearance of punctate structures and was accompanied by SRAS1.1 relocalization to the nucleus, supporting the association of SRAS1.1 with autophagosomes under drought stress (Fig. 6A). These findings demonstrate that drought stress not only regulates SRAS1.1 nuclear-cytoplasmic dynamics but also promotes its association with autophagosomes.

To further explore the involvement of SRAS1.1 in autophagy, we crossed the *35S:GFP-ATG8e* autophagosome marker line with the wild-type and *sras1.1* mutants to generate *35S:GFP-ATG8e* and *35S:GFP-ATG8e/sras1.1* plants. After treatment with 250 mM mannitol, the accumulation of *35S:GFP-ATG8e* labeled punctate autophagosomes significantly increased in wild-type root cells and was even more pronounced in the *sras1.1* background (Fig. 6B,C). These findings demonstrate that the loss-of-function mutation of *sras1.1* enhances autophagosome formation in cells under drought

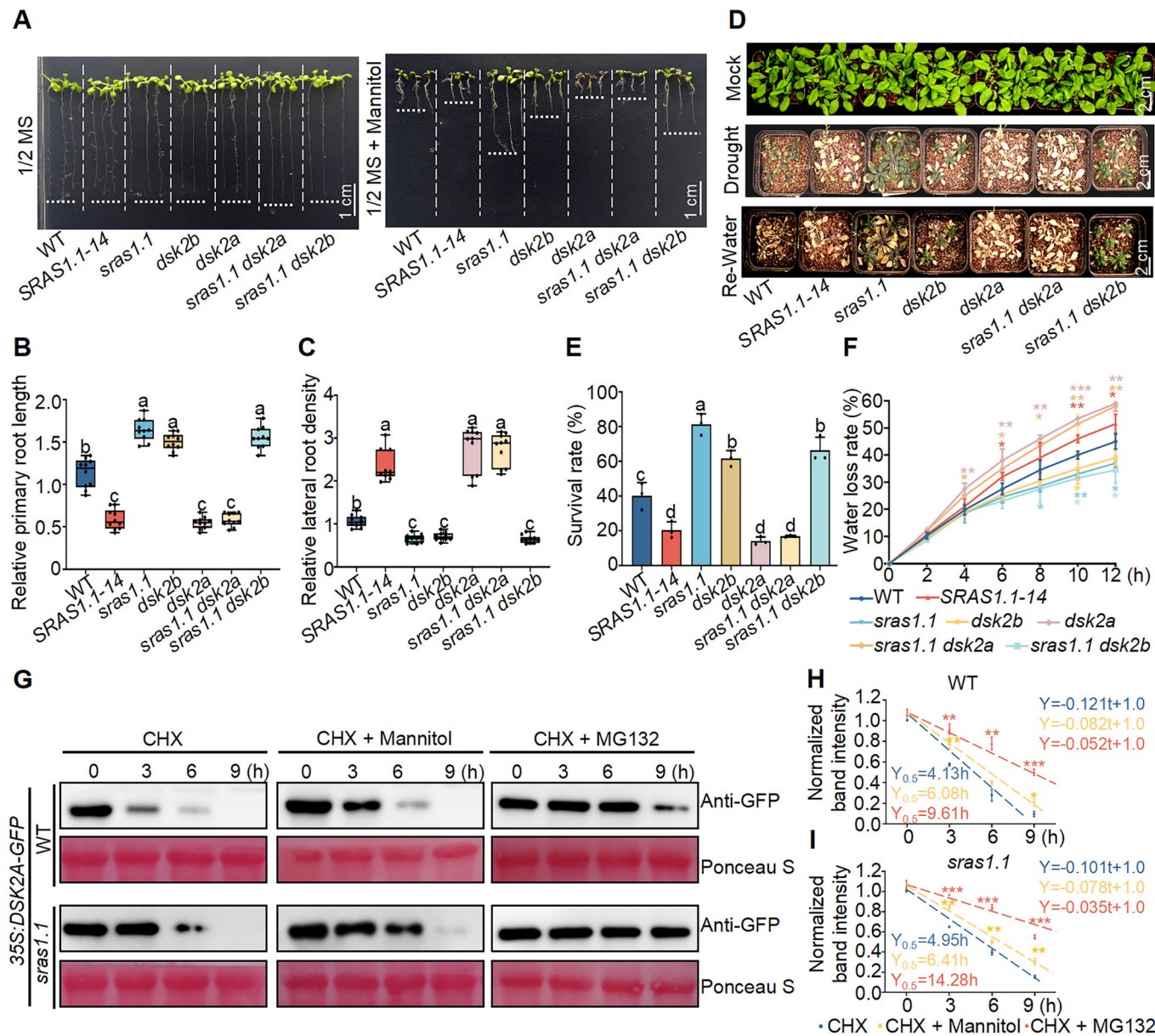

stress. Under drought stress, the expression levels of *ATG* genes, which are reliable markers of autophagy induction (Xiang et al, 2024), were significantly higher in *sras1.1* mutants compared to the wild-type, further confirming the enhanced autophagy activity in the absence of SRAS1.1 (Fig. 6D,E).

Under drought stress, DSK2A mediates the selective autophagy of BES1, a master regulator in the brassinosteroid pathway, to balance plant growth and survival (Nolan et al, 2017). In *sras1.1* mutants, BES1 protein levels were significantly reduced under drought stress (Fig. 6F), which was accompanied by the downregulation of its co-regulators *WRKY54* and *WRKY70* (Fig. 2H,I). These findings suggest that SRAS1.1 enhances BES1 accumulation by negatively regulating DSK2A. This regulatory pathway links SRAS1.1-mediated autophagy and brassinosteroid signaling, contributing to plant growth and drought stress tolerance (Fig. 6G).

## Discussion

Environmental stresses significantly influence the growth and development of both plants and animals by disrupting cellular homeostasis (Cai et al, 2024; Gupta et al, 2020; Wang et al, 2024; Yu et al, 2023). UPS and autophagy are two major cellular degradation mechanisms in eukaryotes, both essential for maintaining cell and tissue homeostasis by removing damaged proteins and organelles in response to external stimuli (Ling et al, 2019; Sun et al, 2021; Xu et al, 2017). Although the two systems operate independently, emerging evidence indicates that E3 ligases play a pivotal role in linking the UPS and autophagy pathways under environmental stress conditions (Wang et al, 2022; Wang et al, 2024; Wei et al, 2021). In animals, E3 ligases regulate autophagy by targeting specific organelles and modulating the core components of the autophagic machinery (Chen et al, 2019; Dikic, 2010; Zhu et al, 2017). The ubiquitin ligase UBE3C regulates the ubiquitination and

Figure 5. **DSK2A is epistatic to SRAS1.1 under drought stress.**

(A) Root growth phenotype of seedlings with the indicated genotypes grown on 1/2 MS medium with or without 250 mM mannitol for 14 days. Scale bars = 1 cm. (B, C) Relative primary root length (B) and lateral root density (C) of seedlings shown in (A), expressed relative to the wild-type (set to 1). Values shown are means ± SD ($n = 3$ biological replicates), with 10 plants analyzed per replicate. (B) P values < 0.0001 (wild-type vs *SRAS1.1-14*), <0.0001 (wild-type vs *sras1.1*), <0.0001 (wild-type vs *dsk2b*), <0.0001 (wild-type vs *dsk2a*), <0.0001 (wild-type vs *sras1.1 dsk2a*), <0.0001 (wild-type vs *sras1.1 dsk2b*), <0.0001 (*SRAS1.1-14* vs *sras1.1*), <0.0001 (*SRAS1.1-14* vs *dsk2b*), 0.9882 (*SRAS1.1-14* vs *dsk2a*), >0.9999 (*SRAS1.1-14* vs *sras1.1 dsk2a*), <0.0001 (*SRAS1.1-14* vs *sras1.1 dsk2b*), 0.12 (*sras1.1* vs *dsk2b*), <0.0001 (*sras1.1* vs *dsk2a*), <0.0001 (*sras1.1* vs *sras1.1 dsk2a*), 0.5569 (*sras1.1* vs *sras1.1 dsk2b*), <0.0001 (*dsk2b* vs *dsk2a*), <0.0001 (*dsk2b* vs *sras1.1 dsk2a*), 0.9732 (*dsk2b* vs *sras1.1 dsk2b*), 0.9942 (*dsk2a* vs *sras1.1 dsk2a*), <0.0001 (*dsk2a* vs *sras1.1 dsk2b*), <0.0001 (*sras1.1 dsk2a* vs *sras1.1 dsk2b*). (The following is the same order). (C) P values < 0.0001, 0.0435, 0.0482, <0.0001, <0.0001, 0.0325, <0.0001, <0.0001, 0.0554, 0.0912, <0.0001, >0.9999, <0.0001, <0.0001, >0.9999, <0.0001, <0.0001, >0.9999, >0.9999, <0.0001, <0.0001. (D) Drought tolerance assay in soil. Wild-type, *SRAS1.1-14*, *sras1.1*, *dsk2b*, *dsk2a*, *sras1.1 dsk2b*, and *sras1.1 dsk2a* plants grown under normal growth conditions for 2 weeks were subjected to drought stress for 16 days and then rewatered for 5 days. Scale bars = 2 cm. (E, F) Survival rates of seedlings following re-watering (E) and water loss rates (F) in detached leaves. Values shown are means ± SD ($n = 3$ biological replicates). (E) P values = 0.0061 (wild-type vs *SRAS1.1-14*), <0.0001 (wild-type vs *sras1.1*), 0.0031 (wild-type vs *dsk2b*), 0.0005 (wild-type vs *dsk2a*), 0.0015 (wild-type vs *sras1.1 dsk2a*), 0.0005 (wild-type vs *sras1.1 dsk2b*), <0.0001 (*SRAS1.1-14* vs *sras1.1*), <0.0001 (*SRAS1.1-14* vs *dsk2b*), 0.7951 (*SRAS1.1-14* vs *dsk2a*), 0.9833 (*SRAS1.1-14* vs *sras1.1 dsk2a*), <0.0001 (*SRAS1.1-14* vs *sras1.1 dsk2b*), 0.0076 (*sras1.1* vs *dsk2b*), <0.0001 (*sras1.1* vs *dsk2a*), <0.0001 (*sras1.1* vs *sras1.1 dsk2a*), 0.0475 (*sras1.1* vs *sras1.1 dsk2b*), <0.0001 (*dsk2b* vs *dsk2a*), <0.0001 (*dsk2b* vs *sras1.1 dsk2a*), 0.9266 (*dsk2b* vs *sras1.1 dsk2b*), 0.9949 (*dsk2a* vs *sras1.1 dsk2a*), <0.0001 (*dsk2a* vs *sras1.1 dsk2b*), <0.0001 (*sras1.1 dsk2a* vs *sras1.1 dsk2b*). (F) T = 2 h: P values = 0.1829 (wild-type vs *SRAS1.1-14*), 0.9304 (wild-type vs *sras1.1*), 0.1731 (wild-type vs *dsk2b*), 0.2222 (wild-type vs *dsk2a*), 0.0539 (wild-type vs *sras1.1 dsk2a*), 0.0657 (wild-type vs *sras1.1 dsk2b*). (The following is the same order). T = 4 h: P values = 0.366, 0.4193, 0.4, 0.0038, 0.031, 0.7369. T = 6 h: P values = 0.0179, 0.2469, 0.2509, 0.0062, 0.016, 0.0748. T = 8 h: P values = 0.2951, 0.0178, 0.0516, 0.0039, 0.0127, 0.109. T = 10 h: P values = 0.0012, 0.0015, 0.0421, <0.0001, 0.0012, 0.0343. T = 12 h: P values = 0.0328, 0.0253, 0.0587, 0.0072, 0.0019, 0.0108. (G) Degradation rates of DSK2A in *35S:DSK2A-GFP*, and *35S:DSK2A-GFP/sras1.1* seedlings treated with or without mannitol and MG132. Seedlings were treated with 200 μM CHX, 200 μM CHX + 250 mM mannitol or 200 μM CHX + 100 μM MG132 for 3, 6, or 9 h. CHX cycloheximide. Similar results were obtained from three independent experiments. Each figure panel displays a representative image obtained from a gel-based assay. (H, I) Linear regressions of the quantified band intensities from (G) by ImageJ representing the degradation rates of DSK2A-GFP in *35S:DSK2A-GFP* (H), and *35S:DSK2A-GFP/sras1.1* (I) with or without mannitol and MG132 treatment. $Y_{0.5}$ represents the time point at which half of the 35S:DSK2A-GFP protein was degraded. All groups were compared with *35S:DSK2A-GFP* separately. Values shown are means ± SD ($n = 3$ biological replicates). (H) T = 3 h: P values = 0.0077 (CHX vs CHX + Mannitol), 0.0014 (CHX vs CHX + MG132). (The following is the same order). T = 6 h: P values = 0.098, 0.0015. T = 9 h: P values = 0.0267, 0.0002. (I) T = 3 h: P values = 0.0016, <0.0001. T = 6 h: P values = 0.0051, <0.0001. T = 9 h: P values = 0.0069, 0.0003. Data information: For (B, C, E) different lowercase letters represent significant differences, as determined by one-way ANOVA in combination with Tukey's comparisons test, ($P < 0.05$). Data in (B, C) are plotted with box–whisker plots: the whiskers represent maximum and minimum values, and boxes represent the upper quartile, median, and lower quartile, dots represent data points. For (F, H, I) asterisks represent significant differences determined by Student's *t* test (*$P < 0.05$; **$P < 0.01$; ***$P < 0.001$). Source data are available online for this figure.

degradation of the class III PI3-kinase complex (VPS34), inhibits the formation and maturation of autophagosomes, controls endoplasmic reticulum quality and cell survival under proteotoxic stress (Chen et al, 2019; Liu et al, 2016). When mitochondria are damaged, the E3 ubiquitin ligase Parkin is recruited by the mitochondrial kinase PINK1 to trigger mitophagy and maintain mitochondrial function (Chen et al, 2019; Geisler et al, 2010; Rasool et al, 2018). Notably, in plants, E3 ubiquitin ligases primarily target autophagy receptors to modulate autophagic pathways. E3 ubiquitin ligase COST1 interacts with autophagy receptor NBR1 to promote ATG8e degradation, enhancing plant drought tolerance (Bao et al, 2020). Here, we demonstrate that SRAS1.1, as an E3 ubiquitin ligase, directly targets and degrades autophagy receptor DSK2A to regulate *ATG* gene expression under drought stress. As a selective autophagy receptor, DSK2A plays a critical role in stress-induced autophagy, and both its subcellular localization and autophagosome association are enhanced in the *sras1.1* mutants under drought conditions (Fig. EV4C,D). These findings highlight E3 ubiquitin ligases as key regulators linking selective autophagy to environmental stress responses in plants (Fig. 6G).

Both DSK2A and DSK2B act as ubiquitination and autophagy receptors (Nolan et al, 2017); however, DSK2A plays a predominant role in plant drought resistance. First, distinct interaction domains likely contribute to this specificity. Analysis of the tertiary structures of DSK2A and DSK2B revealed significant structural differences (Appendix Fig. S6). SRAS1.1 interacts with DSK2A through its RING domain, while its interaction with DSK2B is mediated by the N terminus (Appendix Fig. S7). Further structural studies are needed to pinpoint the molecular determinants underlying SRAS1.1's preference for DSK2A, such as specific regions within the RING domain that enhance DSK2A recognition and degradation. Another explanation involves SRAS1.1's splicing variant, SRAS1.2, which appears to shield

DSK2B from degradation by competing with SRAS1.1. BiFC and LCI assays demonstrated that SRAS1.2 specifically interacts with DSK2B in the cytoplasm (Appendix Fig. S8A–C). Additionally, competitive pull-down assays revealed that higher SRAS1.2 levels reduce SRAS1.1 binding to DSK2B (Appendix Fig. S8D), further supporting its protective role. Importantly, SRAS1.2 maintains a very low expression level under drought stress (Fig. EV5A,B), and phenotypic analyses of *SRAS1.2* overexpressing lines revealed only weak or negligible responses under drought conditions (Fig. EV5C–G; Appendix Fig. S9), further indicating its limited contribution to stress adaptation. We propose that DSK2 plays a crucial role in balancing plant development and stress tolerance: DSK2A primarily regulates drought resistance, while DSK2B supports growth and development. As a ubiquitination receptor, DSK2 contains Ub-like/Ub-associated (UBL/UBA) motifs that facilitate the transport of ubiquitinated proteins to the 26S proteasome for degradation (Farmer et al, 2010; Kaur et al, 2013). However, the differing roles of DSK2A and DSK2B under drought stress warrant further investigation. Future studies should focus on mapping the precise interaction regions of DSK2A and DSK2B with SRAS1.1 and analyzing their structural variations to elucidate the molecular basis of their functional divergence.

The molecular mechanisms by which autophagy enhances drought tolerance, particularly its role in maintaining cellular homeostasis, remain poorly understood (Eckardt et al, 2024). During drought recovery, autophagy resets cellular status by degrading drought-induced proteins (Li et al, 2020; Tang & Bassham, 2021; Xiang et al, 2024). Emerging evidence suggests that autophagy enhances drought resistance through interactions with hormone signaling networks, particularly brassinosteroid signaling (Nolan et al, 2017; Xiang et al, 2024). For example, the maize cell autophagy receptor ZmNBR1 facilitates the degradation of the brassinosteroid receptor BRI1a, thereby reducing

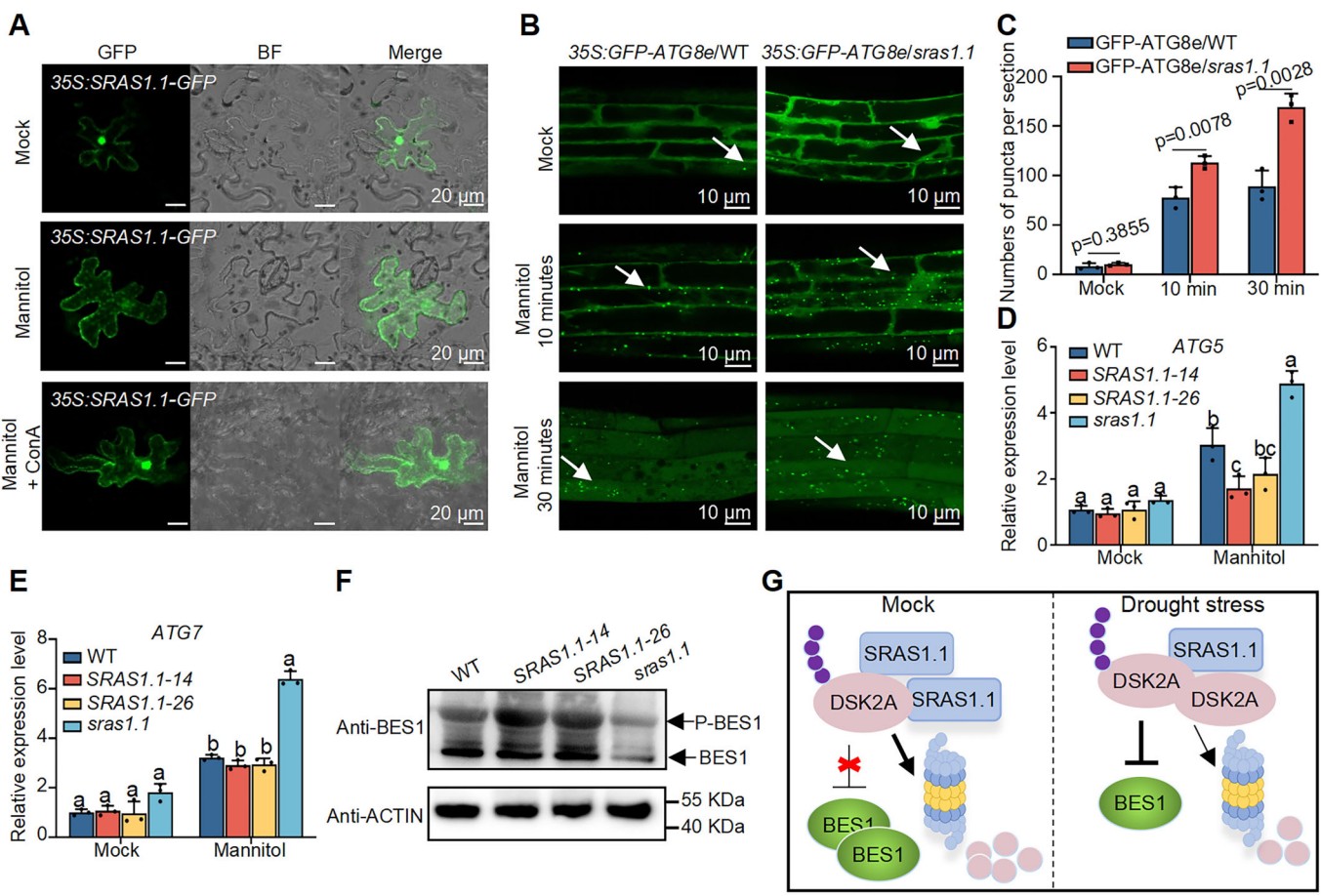

**Figure 6. SRAS1.1 activates drought stress-induced autophagy.**

(A) Localization of transiently expressed *35S:SRAS1.1-GFP*. Confocal microscopy of *N. benthamiana* transfected with *35S:SRAS1.1-GFP*, grown under light for 2 days, then placed in liquid medium supplemented with 250 mM mannitol and 10 μM ConA for 6 h, and then observed by confocal laser scanning microscopy. The arrows indicate SRAS1.1-GFP spots. ConA, concanamycin A. Scale bars = 20 μm. (B) Confocal analysis of *GFP-ATG8e/WT*, and *GFP-ATG8e/sras1.1* lines. Five-day-old seedlings were exposed to 250 mM mannitol liquid medium and then visualized by confocal laser scanning microscopy. The arrows indicate autophagic bodies. Scale bars = 10 μm. (C) Numbers of puncta per section in the root cells of the *GFP-ATG8e/WT*, and *GFP–ATG8e/sras1.1* in (B). Values shown are means ± SD (*n* = 3 biological replicates), with 10 plants analyzed per replicate. Significance was determined using Student's *t* test. Mock: *P* values = 0.3855 (*GFP-ATG8e/WT* vs *GFP-ATG8e/sras1.1*). 10 min: *P* values = 0.0078. 30 min: *P* values = 0.0028. (D, E) Relative expression of autophagy-related genes *ATG5* (D) and *ATG7* (E) gene by qRT-PCR analysis normalized to *UBQ10* levels in 14-day-old *Arabidopsis* grown for 16 days on 1/2 MS with or without 250 mM mannitol. Values shown are means ± SD (*n* = 3 biological replicates). (D) Mock: *P* values = 0.8339 (wild-type vs *SRAS1.1-14*), >0.9999 (wild-type vs *SRAS1.1-26*), 0.2346 (wild-type vs *sras1.1*), 0.8558 (*SRAS1.1-14* vs *SRAS1.1-26*), 0.0749 (*SRAS1.1-14* vs *sras1.1*), 0.2236 (*SRAS1.1-26* vs *sras1.1*). (The following is the same order). Mannitol: *P* values = 0.0258, 0.1442, 0.0037, 0.6271, <0.0001, 0.0003. (E) Mock: *P* values = 0.9928, 0.9986, 0.0694, 0.9731, 0.1013, 0.0558. Mannitol: *P* values = 0.3984, 0.469, <0.0001, 0.9988, <0.0001, <0.0001. (F) Western blot analysis of BES1 protein levels in wild-type, *SRAS1.1-14*, *SRAS1.1-26*, and *sras1.1* mutants. ACTIN was used as a loading control. Similar results were obtained from three independent experiments. Each figure panel displays a representative image obtained from a gel-based assay. (G) Model in which SRAS1.1 promotes the degradation of DSK2A to ensure the stability of BES1 protein. Drought stress leads to the accumulation of DSK2A, which in turn promotes the degradation of BES1, balancing the growth and development of plants. Data information: For (C) significance was determined using Student's *t* test. For (D, E) different lowercase letters represent significant differences, as determined by one-way ANOVA in combination with Tukey's multiple comparisons test (*P* < 0.05). Source data are available online for this figure.

water loss and enhancing drought stress resistance (Xiang et al, 2024). Under drought stress, GSK3-like kinase BIN2 phosphorylates the autophagy receptor DSK2, promoting the degradation of BES1 and enhancing plant survival. Additionally, BIN2 phosphorylates and stabilizes the AP2/ERF transcription factor TINY, promoting ABA-induced stomatal closure and further enhancing drought resistance (Gupta et al, 2020; Nolan et al, 2017). Consistent with these findings, brassinosteroid signaling-related genes are differentially expressed in the *sras1.1* mutants under drought stress. The *sras1.1* mutants also exhibit increased autophagosome accumulation, primary root length and

reduced stomatal aperture, indicative of altered brassinosteroid signaling. Future studies should focus on uncovering the precise mechanisms by which autophagy contributes to drought resistance and its interplay with BR signaling. SRAS1 belongs to the *Arabidopsis* Tóxicos en Levadura (ATL) subfamily of RING-type E3 ubiquitin ligases, which is highly conserved across diverse plant species (Zhou et al, 2021). Despite its established role in drought tolerance, particularly in crops, the molecular mechanisms by which autophagy mediates survival under stress remain poorly characterized. Understanding how ubiquitination regulates autophagy receptors, particularly in

autophagosome formation and maturation under abiotic stress, is critical for enhancing plant resilience.

This study establishes SRAS1.1 as a pivotal link between the UPS and autophagy, revealing a novel regulatory pathway critical for plant drought tolerance. Future research should explore upstream regulators and leverage genetic engineering to enhance crop resilience. Discovering additional E3 ligases involved in autophagy could further uncover regulatory dimensions, driving advancements in sustainable agriculture.

# Methods

### Reagents and tools table

| Reagent/resource | Reference or source | Identifier or catalog number |
|---|---|---|
| **Experimental models** | | |
| *Arabidopsis*: Col-0 | Widely distributed | N/A |
| *Arabidopsis*: proSRAS1.1:GUS | This study | N/A |
| *Arabidopsis*: SRAS1.1-14 | Zhou et al, 2021 | N/A |
| *Arabidopsis*: SRAS1.1-26 | Zhou et al, 2021 | N/A |
| *Arabidopsis*: sras1.1 | Zhou et al, 2021 | SALK_034426 |
| *Arabidopsis*: dsk2a | Arashare | SALK_058067 |
| *Arabidopsis*: dsk2b | Arashare | SALK_152333C |
| *Arabidopsis*: sras1.1 dsk2a | This study | N/A |
| *Arabidopsis*: sras1.1 dsk2b | This study | N/A |
| *Arabidopsis*: 35S:DSK2A-GFP | This study | N/A |
| *Arabidopsis*: 35S:DSK2A-GFP/ sras1.1 | This study | N/A |
| *Arabidopsis*: 35S:DSK2B-GFP/ sras1.1 | This study | N/A |
| *Arabidopsis*: 35S:SRAS1.1-GFP | This study | N/A |
| *Arabidopsis*: 35S:GFP-ATG8e | This study | N/A |
| *Arabidopsis*: proSRAS1.1:SRAS1.1/ sras1.1 | This study | N/A |
| *Arabidopsis*: SRAS1.2-1 | Zhou et al, 2021 | N/A |
| *Arabidopsis*: SRAS1.2-4 | Zhou et al, 2021 | N/A |
| **Recombinant DNA** | | |
| *proSRAS1.1:GUS* | This study | N/A |
| *SRAS1.1-His* | Zhou et al, 2021 | N/A |
| *SRAS1.1-BD* | This study | N/A |
| *DSK2A-AD* | This study | N/A |
| *DSK2B-AD* | This study | N/A |
| *SRAS1.1$_{\Delta RING}$-BD* | Zhou et al, 2021 | N/A |
| *SRAS1.1$_{(1-67)}$-BD* | Zhou et al, 2021 | N/A |

| Reagent/resource | Reference or source | Identifier or catalog number |
|---|---|---|
| *SRAS1.1$_{(1-19)}$-BD* | Zhou et al, 2021 | N/A |
| *SRAS1.1$_{RING}$-BD* | Zhou et al, 2021 | N/A |
| *SRAS1.1-cLUC* | Zhou et al, 2021 | N/A |
| *nLUC-DSK2A* | This study | N/A |
| *nLUC-DSK2B* | This study | N/A |
| *35S:SRAS1.1-GFP* | Zhou et al, 2021 | N/A |
| *DSK2A-HA* | This study | N/A |
| *DSK2B-HA* | This study | N/A |
| *SRAS1.1-cYFP* | Zhou et al, 2021 | N/A |
| *nYFP-DSK2A* | This study | N/A |
| *nYFP-DSK2B* | This study | N/A |
| *GST-SRAS1.1* | Zhou et al, 2021 | N/A |
| *SRAS1.2-GST* | Zhou et al, 2021 | N/A |
| *35S:DSK2A-GFP* | This study | N/A |
| *35S:DSK2B-GFP* | This study | N/A |
| *35S:GFP-ATG8e* | This study | N/A |
| *DSK2A-His* | This study | N/A |
| *DSK2B-His* | This study | N/A |
| *DSK2A$^{K68R}$-His* | This study | N/A |
| *DSK2A$^{K29R}$-His* | This study | N/A |
| *SRAS1.2-cYFP* | Zhou et al, 2021 | N/A |
| *SRAS1.2-cLUC* | Zhou et al, 2021 | N/A |
| *GST-SRAS1.2* | Zhou et al, 2021 | N/A |
| *nYFP-CSN5A* | Zhou et al, 2021 | N/A |
| *nLUC-CSN5A* | Zhou et al, 2021 | N/A |
| *nYFP-ABI5* | Zhou et al, 2021 | N/A |
| *nLUC-ABI5* | Zhou et al, 2021 | N/A |
| **Antibodies** | | |
| Mouse monoclonal anti-GFP | EasyBio | Cat#BE2001 |
| Mouse monoclonal anti-HA | Solarbio | Cat#K200003M |
| Mouse monoclonal anti-GST | Solarbio | Cat#K200006M |
| Mouse monoclonal anti-His | Solarbio | Cat#K200060M |
| Mouse monoclonal anti-ACTIN | Solarbio | Cat#K200058M |
| Rabbit polyclonal anti-MYC | ABclonal | Cat#AE010 |
| Rabbit polyclonal anti-DSK2 | This study | N/A |
| Rabbit polyclonal anti-BES1 | This study | N/A |

| Reagent/resource | Reference or source | Identifier or catalog number |
|---|---|---|
| Goat anti-Rabbit IgG/HRP | Solarbio | Cat#SE134 |
| Goat anti-Mouse IgG/HRP | Solarbio | Cat#SE131 |
| **Oligonucleotides and other sequence-based reagents** | | |
| Primers for the generation and identification of transgenic *Arabidopsis* | This study | See Table EV1 for details |
| qRT-PCR primers | This study | See Table EV1 for details |
| **Chemicals, enzymes and other reagents** | | |
| Kanamycin Sulfate (1:1000) | AMRESCO | Cat#7019B276 |
| Ampicillin Sodium Salt (1:1000) | AMRESCO | Cat#1190C267 |
| Rifampicin (1:1000) | Solarbio | Cat#13292-46-1 |
| Cycloheximide (CHX) | Aladdin | Cat#66-81-9 |
| Protease Inhibitor MG132 | Sigma-Aldrich | Cat#M7449 |
| Concanamycin A | Sigma-Aldrich | Cat#C9705 |
| Mannitol | Sigma-Aldrich | Cat#M4125 |
| Sodium chloride | Sinopharm | Cat#10019318 |
| Sucrose | Sinopharm | Cat#10021418 |
| IPTG | Solarbio | Cat#11020 |
| MES | Sigma-Aldrich | Cat#M3671 |
| AS | Sigma-Aldrich | Cat#D134406-1G |
| Tris-HCl | Solarbio | Cat#T8230 |
| Triton X-100 | Solarbio | Cat#T8200 |
| EDTA | Solarbio | Cat#E8030 |
| $MgCl_2$ | Sinopharm | Cat#RD11600770103 |
| DTT | Solarbio | Cat#D8220 |
| Murashige and skoog's medium | Duchefa Biochemie | Cat#P14187.01 |
| SD-Trp-Leu | Solarbio | Cat#S6110 |
| SD-Trp-Leu-His-Ade | Solarbio | Cat#S6120 |
| Agar Powder | Solarbio | Cat#A8190 |
| Agarose | Solarbio | Cat#A8201 |
| Trizol | Thermo Fisher Scientific | Cat#1559601 |
| Trypsin | Roche | Cat#03708969001 |
| Ponceau S Solution | Solarbio | Cat#P0012 |
| Fluorescein diacetate (FDA) | Sigma-Aldrich | Cat#F7378 |
| Alexander | Solarbio | Cat#G3050 |
| 4-methylumbelliferone | Sigma-Aldrich | Cat#M1381 |
| Immobilon-P Membran, PVDF, 0.45 mm | Millipore | Cat#IPVH00010 |
| Tween 20 | Sigma-Aldrich | Cat#P2287 |
| 10×TBST Buffer Western | NCM Biotech | Cat#WB20500 |
| PMSF | Solarbio | Cat#P0100 |
| D-luciferin potassium salt | Solarbio | Cat#IL2330 |
| Glutathione beads | Smart-lifesciences | Cat#SA008010 |

| Reagent/resource | Reference or source | Identifier or catalog number |
|---|---|---|
| Protein loading buffer | NCM Biotech | Cat#WB2001 |
| Prestained Protein Marker | Vazyme | Cat#MP102 |
| DNA Marker 5000 | Accurate Biology | Cat#AG11906 |
| SYBR Green Real-time PCR Master Mix | Accurate Biology | Cat#AG11759 |
| RNAsimple extraction kit | TIANGEN | Cat#DP441 |
| Transcriptor First Strand cDNA Synthesis Kit | Thermo Fisher Scientific | Cat#K1682 |
| GUS Stain Kit | Solarbio | Cat#G3060 |
| 2×Flash PCR MasterMix (Dye) | CWBIO | Cat#CW3009H |
| His tag protein purification kit | Beyotime Biotechnology | Cat#P2245S |
| GST tag protein purification kit | Beyotime Biotechnology | Cat#P2260S |
| Express Cast PAGE | NCM Biotech | Cat#P2013 |
| 2×Phanta Flash Master Mix | Accurate Biology | Cat#AG12202 |
| 5×Evo M-MLV RT Reaction Mix | Accurate Biology | Cat#AG11728 |
| T4 DNA ligase | NEB | Cat#1151505 |
| QuickPure Plasmid Mini Kit | CWBIO | Cat#CW2619M |
| pEASY-Uni Seamless Cloning and Assembly Kit | TransGen Biotech | Cat#CU101 |
| **Software** | | |
| ImageJ | National Institutes of Health | https://ImageJ.nih.gov/ij/ |
| Office Excel Power Point 2016 | Microsoft | N/A |
| Graphpad PRISM8.0 | GraphPad Software Inc. | https://www.graphpad.com |
| Uniprot | UniProt Consortium | https://www.uniprot.org/ |
| ZEN 3.3 | Carl Zeiss Microscopy GmbH | https://www.micro-shop.zeiss.com/en/de/softwarefinder/ |
| DNAMAN Version 9 | Lynnon Corporation | https://www.lynnon.com/dnaman.html |
| **Other** | | |
| *Echerichia coli (E. coli)* strain BL21(DE3) | Novagen | Cat#C504-02 |
| *E.coli* (DH5α) | Lab stock | N/A |
| *Agrobacterium tumefaciens* (strain GV3101) | Lab stock | N/A |
| Yeast Strain (Y2HGold) | Lab stock | N/A |

## Plant materials and growth conditions

*Arabidopsis* plants used in this study were of the Columbia-0 (Col-0) background. Descriptions of the *SRAS1.1-14*, *SRAS1.1-26*, *SRAS1.2-1*, *SRAS1.2-4*, and *sras1.1* genotypes were provided in

the literature (Zhou et al, 2021). The *dsk2a* (SALK_058067) and *dsk2b* (SALK_152333C) mutants were obtained from the *AraShare* (www.arashare.cn). All mutants were confirmed by PCR using primers listed in Table EV1. The *sras1.1* genotype was crossed with *dsk2a* and *dsk2b* to generate the *sras1.1 dsk2a* and *sras1.1 dsk2b* lines. For the *proSRAS1.1:GUS* plants, a 1-kb genomic fragment upstream of the start codon was fused to the *pBI121-GUS* vector and transformed into Col-0 via the floral dip method. $T_1$ positive plants were identified by PCR and sequencing, and homozygous transgenic plants were obtained in the $T_2$ generation. For the *35S:GFP-ATG8e*, *35S:DSK2A-GFP*, *35S:DSK2B-GFP* and *35S:SRAS1.1-GFP* constructs, the coding sequences (CDS) were cloned into pMDC43-GFP and pROK II-GFP, respectively. The constructs were introduced into Col-0 and sras1.1 backgrounds. To generate *SRAS1.1* complementation line (COM1), DNA fragments including 1 kb upstream of *SRAS1.1* and the full-length *SRAS1.1* sequence were amplified from Col-0 genomic DNA, and then cloned into the vector pBI121. All produced plasmids were introduced into the *Agrobacterium tumefaciens* strain GV3101, and then transformed into *Arabidopsis thaliana* plants via the floral dip method (Li et al, 2021). In brief, $T_0$ transgenic seeds were selected on 1/2 Murashige-Skoog (1/2 MS) medium containing 44 mg/L kanamycin. Positive seedlings were transferred to soil to generate $T_1$ plants. $T_1$ plants were used to assess protein and transcript levels, with the primers listed in Table EV1.

*Arabidopsis* seeds were surface sterilized and sown on 1/2 MS medium containing 1.5% agar. Following a stratification period of 3 days at 4 °C, the seeds were positioned either horizontally or vertically in a light incubator. All *Arabidopsis* and *Nicotiana benthamiana* materials were grown under long-day conditions (16 h light/8 h dark) in a plant growth chamber (PERCIVAL, Thermo Fisher, USA) at 120 µmol m$^{-2}$ s$^{-1}$ (white fluorescent bulbs), and 65% relative humidity. *Arabidopsis* was grown at 22 °C and *Nicotiana benthamiana* at 26 °C.

## Physiological experiments

To measure root growth, seedlings were cultivated vertically on 1/2 MS medium, with or without 250 mM mannitol. Root lengths were quantified using the ImageJ software (https://imagej.nih.gov/ij). For soil drought assessments, 7-day-old seedlings were transferred to soil and allowed to grow for an additional 7 days. After thoroughly watering each pot, the plants were subjected to a 16-day period before photographs were taken. They were then rewatered and allowed to recover for 5 days. Equal amounts of soil were used for each pot, and pot positions were randomized every 2 days. To evaluate water loss from detached leaves, rosette leaves from plants grown under standard greenhouse conditions for 3 weeks were excised and placed on weighing paper. Detached rosette leaves were weighed every 2 h over 12 h to assess water loss. Water loss rates were expressed as a percentage of the initial fresh weight, with at least three independent experiments conducted, each involving 10 leaves per genotype. For stomatal observation, complete rosette leaves from plants subjected to drought stress for 2 weeks were collected. The leaf stomata were examined using scanning electron microscopy (Carl Zeiss Microscopy GmbH, Jena, Germany). The length, width, and aperture of 20 stomata from various locations on the same sample were measured using ImageJ, with the experiment conducted across three biological replicates. Regarding the statistics

of rosette leaf number, the total count of leaves was recorded when the primary inflorescence reached approximately 5 cm in height. Flowering time and the number of rosette leaves were represented using box plots, with the median indicated within the plots. All physiological experiments were conducted as previously described, with some modifications (Chen et al, 2021b; Liu et al, 2023b).

## RNA extraction and qRT-PCR

To verify the effects of drought on *SRAS1.1* transcript levels, 7-day-old wild-type seedlings were treated with 250 mM mannitol for durations of 1, 3, 6, 9, and 12 h. Total RNA was extracted using the RNAsimple extraction kit (TIANGEN). Following the determination of RNA concentration, 1 µg of RNA was reverse-transcribed into first-strand complementary DNA (cDNA) in accordance with the instructions provided by the Transcriptor First Strand cDNA Synthesis Kit (Thermo Scientific). Quantitative real-time PCR (qRT-PCR) was conducted on a MyiQ real-time PCR detection system using the SYBR green real-time PCR master mix (Accurate Biology). *UBQ10* was used as the internal reference gene. Primer sequences used for qRT-PCR are listed in Table EV1 (see Reagents and Tools Table).

## GUS staining and GUS activity analysis

For GUS staining, 8-day-old *ProSRAS1.1:GUS* seedlings were transferred to 1/2 MS medium containing 250 mM mannitol for 2 h of vertical growth. Following this, *Arabidopsis* seedlings were placed in GUS staining solution [2 mM X-Gluc (5-bromo-4-chloro-3-indolyl glucuronide), 2 mM $K_3[Fe(CN)_6]$, 100 mM sodium phosphate buffer (pH 7.2), 2 mM $K_4Fe(CN)_6$, 0.1% Triton X-100, and 10 mM $Na_2EDTA$], and incubated overnight at 37 °C. The samples were then destained in 70% ethanol at 60 °C and visualized using an Axioskop2 Plus microscope (Carl Zeiss MicroImaging GmbH, Göttingen, Germany). For the assessment of GUS enzyme activity, ~0.2 g of fresh tissue was ground in liquid nitrogen and transferred to a 2 mL microcentrifuge tube. GUS activity was measured using 4-methylumbelliferyl-β-d-glucuronide as a substrate with an F-4500 fluorescence spectrofluorometer (Hitachi, Tokyo, Japan). Standard curves were prepared with 4-methylumbelliferone. Average GUS activity was obtained from 5 independent transformants, and each assay was repeated 5 times. The detailed methodology is referenced (Liu et al, 2020).

## Pollen staining

Flowers were examined and photographed using an Axioskop2 Plus microscope (Carl Zeiss MicroImaging GmbH, Göttingen, Germany). To observe the morphology of pollen grains, they were stained with Alexander stain. Pollen grains were placed on a glass slide to which Alexander stain was added. After 1 min, the excess stain was absorbed with absorbent paper, and the samples were then observed. Pollen viability was examined using fluorescein diacetate (FDA) staining as described in a previous study (Lei et al, 2017). Pollen was placed in a 1.5 mL centrifuge tube, washed once with saline, and treated with 100 µg/mL FDA. After a 5-min incubation in the dark at room temperature, the pollen was washed 1–2 times with saline, resuspended in distilled water, and observed under a microscope after smear preparation. Images of the Alexander-stained and FDA-stained

materials were captured under bright light and GFP light respectively, and then transfected cells were examined using an LSM 880 confocal laser scanning microscope (Carl Zeiss MicroImaging GmbH, Göttingen, Germany). For scanning electron microscopy (SEM), fresh mature pollen grains were directly coated with gold and analyzed using a Zeiss EVO scanning electron microscope (Carl Zeiss MicroImaging GmbH, Göttingen, Germany).

## RNA-seq analysis

RNA-seq analysis were performed as previously described (Xu et al, 2023). Total RNA was extracted from 10-day-old wild-type, *sras1.1* mutants, and *SRAS1.1-14* overexpressing lines, all of which were treated with 250 mM mannitol for 6 h. The extracted RNA was subsequently utilized for Illumina HiSeq deep sequencing (HiSeq 2500; Illumina, San Diego, CA, USA). The sequencing experiments and data analysis were conducted by Beijing Bio-Bio Technology Co., Ltd.

## LC-MS/MS assay

The liquid chromatography tandem mass spectrometry (LC-MS/MS) assay was conducted as previously described (Zhang et al, 2023). The SRAS1.1-His fusion protein was incubated with total protein extracted from 7-day-old wild-type plants with and without a 250 mM mannitol treatment, the seedlings were ground into powder with liquid nitrogen; 5 g of powder was immediately transferred to a precooled 50 mL centrifuge tube and resuspended in 4 mL of precooled lysis buffer [50 mM Tris-HCl (pH 7.5), 150 mM NaCl, 1% (v/v) Triton X-100, 5 mM EDTA, and a protease inhibitor cocktail tablet]. The samples were incubated at 4 °C for 30 min, followed by centrifugation at 14,000 × *g* for 10 min at 4 °C. The supernatant was then transferred to a new 50 mL centrifuge tube, to which 50 μL of agarose beads coupled with anti-His antibody (1:500 Solarbio) were added and slowly rotated overnight at 4 °C, followed by three washes with 1× PBS buffer, each lasting 10 min. Beads were collected by centrifugation at 1000 × *g* for 10 min at 4 °C, washed three times with lysis buffer, resuspended in 5× SDS protein loading buffer, and boiled at 95 °C for 10 min. The gel sample containing SRAS1.1-His interacting proteins was excised, and the proteins within the gel slice were digested with trypsin at an enzyme-substrate ratio of 1:50 at 37 °C overnight. Finally, the LC-MS/MS assay was performed.

## Y2H assay

The CDS of *SRAS1.1* and its various truncations were cloned into the pGBKT7 vector, while the CDS of *DSK2A* and *DSK2B* were cloned into the pGADT7 vector. These plasmids were co-transformed into the yeast strain Gold. Following transfection, the yeast cells were plated on SD/-Leu-Trp (DDO) medium and incubated at 30 °C for 2 days. Subsequently, yeast spots were cultured in SD/-Leu-Trp liquid medium for one day before being transferred to both DDO solid medium and SD/-Ade-His-Leu-Trp (QDO) solid medium, where they were incubated for an additional 3 days. The primers used in this study are listed in Table EV1 (see Reagents and Tools Table).

## LCI assay

For the LCI assay, the CDS of *SRAS1.1* and *SRAS1.2* was cloned into the pCAMBIA1300-cLUC vector to create the SRAS1.1-cLUC and SRAS1.2-cLUC constructs, while the CDS of *DSK2A*, *DSK2B*, *ABI5* and *CSN5A* was cloned into the pCAMBIA1300-nLUC vector to produce the nLUC-DSK2A, nLUC-DSK2B, nLUC-ABI5 and nLUC-CSN5A constructs. The plasmids and empty vectors were individually transformed into *Agrobacterium tumefaciens* strain GV3101 and infiltrated into four different regions of 30-day-old *N. benthamiana* leaves in the following combinations: DSK2A-nLUC + cLUC, DSK2B-nLUC + cLUC, DSK2A-nLUC + SRAS1.1-cLUC, and DSK2B-nLUC + SRAS1.1-cLUC, with each combination infiltrated into distinct leaf regions. After growth in darkness for 12 h, the infiltrated *N. benthamiana* plants were transferred to light and grown at 26 °C for 48 h. Prior to the assay, 0.2 mM D-luciferin potassium salt was sprayed onto the leaves. The luciferase (LUC) signal was captured using a cooled charge-coupled device camera (Lumina II system; PerkinElmer, Waltham, MA, USA). Primers used in the LCI assays are listed in Table EV1 (see Reagents and Tools Table).

## BiFC assay

Bimolecular fluorescence complementation assays (BiFC) were performed as previously described (Li et al, 2021). Briefly, the full-length CDS of *DSK2A*, *DSK2B*, *ABI5* and *CSN5A* was cloned into the pSPYNE-35S vector to generate DSK2A-nYFP, DSK2B-nYFP, ABI5-nYFP and CSN5A-nYFP constructs. The full-length CDS of *SRAS1.1* and *SRAS1.2* (excluding their stop codons) were also cloned into pSPYCE-35S, yielding SRAS1.1-cYFP and SRAS1.2-cYFP constructs. The following plasmid combinations were co-transformed into *N. benthamiana* leaves: nYFP-DSK2A + SRAS1.1-cYFP, nYFP-DSK2B + SRAS1.1-cYFP, nYFP-ABI5 + SRAS1.1-cYFP, nYFP-CSN5A + SRAS1.1-cYFP, nYFP-DSK2A + cYFP, nYFP-DSK2B + cYFP, nYFP-DSK2A + SRAS1.2-cYFP, nYFP-DSK2B + SRAS1.2-cYFP, and nYFP + SRAS1.2-cYFP. After 48 h of expression, leaves were examined using an LSM 880 confocal laser scanning microscope (Carl Zeiss MicroImaging GmbH, Göttingen, Germany). The primer sequences are listed in Table EV1 (see Reagents and Tools Table).

## Co-IP assay

The CDS of *SRAS1.1* was cloned into the pROKII-GFP vector to generate the *35S:SRAS1.1-GFP* construct (Zhou et al, 2021). Similarly, the CDS of *DSK2A* and *DSK2B* was cloned into the pZP211-3HA vector, resulting in the *35S:DSK2A-HA* and *35S:DSK2B-HA* constructs. Subsequently, SRAS1.1-GFP/DSK2A-HA and SRAS1.1-GFP/DSK2B-HA were co-infiltrated into *N. benthamiana* leaves. After ~72 h, the leaves were harvested for protein extraction. Anti-GFP antibodies coupled to magnetic beads were mixed with the protein samples and incubated at 4 °C for 5–6 h. The captured proteins were then separated by SDS-PAGE. Anti-GFP antibodies (1:1000 dilution, ABclonal) and anti-HA antibodies (1:1000 dilution, Solarbio) were used to detect SRAS1.1-GFP, DSK2A-HA, and DSK2B-HA, respectively. The primers used for Co-IP are listed in Table EV1 (see Reagents and Tools Table).

For detailed methodological references, consult the relevant literature (Li et al, 2021).

## In vitro ubiquitination assay

The in vitro ubiquitination assay was performed as previously described with minor modifications (Chen et al, 2021b; Jiao et al, 2024). *SRAS1.1* was cloned into the pGEX-4T-3 vector (Zhou et al, 2021), while DSK2A and DSK2B were individually fused into the pET-30a vector. Recombinant proteins were expressed in the Escherichia coli strain BL21 (DE3) and subsequently purified using glutathione-sepharose. A total of 100 ng of wheat (Triticum aestivum) E1, 200 ng of purified E2, 5 μg of Myc-tagged ubiquitin, and 1 mg each of purified GST-SRAS1.1, DSK2A-His, and DSK2B-His were added to 30 μL of ubiquitination reaction buffer (50 mM Tris-HCl, pH 7.5; 2 mM ATP; 5 mM $MgCl_2$; 2 mM DTT). After incubating for 24 h at 30 °C, the reactions were terminated by the addition of 5× loading buffer, and the samples were boiled at 100 °C for 5 min. The products were then separated by electrophoresis on a 15% SDS polyacrylamide gel and detected using anti-GST (1:1000 dilution, Solarbio) and anti-Myc (1:1000 dilution, ABclonal) antibodies via western blotting.

## In vivo ubiquitination assay

The in vivo ubiquitination assay was carried out as described previously with some modifications (Zhou et al, 2021). The CDS of *UBQ10* was amplified and cloned into pCAMBIA1300-MYC to generate ubiquitin-MYC. Ubiquitin-MYC was transferred into the protoplasts of *35S:DSK2A-GFP*, *35S:DSK2B-GFP*, *35S:DSK2A-GFP/ sras1.1* and *35S:DSK2B-GFP/sras1.1*, respectively, followed by incubation for 12 h. Then 500 μL of protoplasts were transferred to 2 mL centrifuge tubes, incubated at room temperature for 20 min, and centrifuged at 200 g for 2 min. After removing the supernatant, SDS loading buffer was added to terminate the reaction, and the sample was stored at −80 °C as an input sample. A 5 mL aliquot of protoplasts was transferred to a 10 mL centrifuge tube, and 200 μL lysis buffer was added immediately to the sample, followed by incubation at 4 °C for 30 min, during which the protoplasts were gently shaken every 10 min. The sample was centrifuged at 4 °C at 20,000 × g for 10 min, and the supernatant was transferred to a pre-cooled 2 mL centrifuge tube. A 5 μL aliquot of anti-GFP antibody-coupled beads was placed in a new 2 mL centrifuge tube. The beads were washed three times in dilution buffer, added to the sample, and incubated at 4 °C for 2 h. The beads were washed three times in dilution buffer with 1% (v/v) Triton X-100 and once in 50 mM Tris-HCl (pH 7.5). Finally, the reaction was terminated by adding SDS loading buffer, and anti-MYC antibody was used to detect the ubiquitination levels of DSK2A-GFP and DSK2B-GFP.

## Cell-free degradation assay

Cell-free degradation assays were conducted as previously described (Zhou et al, 2021). Briefly, total proteins from 7-day-old wild-type, SRAS1.1-14, and sras1.1 mutant seedlings were extracted using a native protein extraction buffer [5 mM DTT, 10 mM NaCl, 25 mM Tris-HCl (pH 7.5), 10 mM $MgCl_2$, 10 mM ATP, and 4 mM phenylmethylsulfonyl fluoride]. A total of 200 ng

of purified DSK2A-His, DSK2A$^{K68R}$-His, DSK2A$^{K29R}$-His, and DSK2B-His proteins purified from the E. coli strain BL21 (DE3) were combined with 200 μL of crude protein extracts from different genotypes. Subsequently, 1 mM ATP was added, and the samples were incubated at 22 °C for 0, 1, 2, and 3 h. The reaction was terminated by the addition of 5 × SDS loading buffer. Following this, the samples were boiled and analyzed using anti-His antibodies. After quantifying the protein signal intensity with ImageJ, the linear regression equation and half-life of DSK2A-His and DSK2B-His in various protein extracts were calculated as described previously (Yu et al, 2019).

## Subcellular localization assay

To determine the effect of drought on SRAS1.1 localization, the *SRAS1.1* CDS was fused to the coding region of the pROKII-GFP vector, driven by the *CaMV35S* promoter, resulting in the *35S:SRAS1.1-GFP* construct. This construct was infiltrated into the leaves of *N. benthamiana* plants, allowing expression for three days. The infiltrated *N. benthamiana* plants were subsequently treated with 250 mM mannitol and 10 μM concanamycin A (ConA) for 3 h. For imaging of *35S:SRAS1.1-GFP* and *35S:DSK2A-GFP* signals, homozygous *35S:SRAS1.1-GFP* and *35S:DSK2A-GFP* lines were grown for 5 days in light and then transferred to mannitol and ConA plates for 1-hour treatments. To examine changes in autophagy between wild-type and *sras1.1* mutants, stable lines of *35S:GFP-ATG8e* and *35S:GFP-ATG8e/sras1.1* were generated, and GFP fluorescence was assessed in the roots of 5-day-old transgenic plants. All images were acquired using an LSM880 Airyscan inverted confocal laser scanning microscope (Carl Zeiss Micro-Imaging GmbH, Göttingen, Germany) with either a 20× or 40× water-corrected objective. The green fluorescent protein (GFP) was excited with an argon laser at 488 nm. The signal intensity of *35S:SRAS1.1-GFP* in the nucleus, as well as the total cell signal intensity, were quantified using ImageJ to calculate the percentage of SRAS1.1 signal localized in the nucleus.

## Quantification and statistical analysis

A one-way analysis of variance (ANOVA) followed by Tukey's honestly significant difference (HSD) test was utilized to obtain $P$ values, with different letters or asterisks above the bars indicating significant differences at $P < 0.05$. The Student's $t$ test was employed to evaluate the differences between the two groups of data at a specific time point (*$P < 0.05$; **$P < 0.01$; ***$P < 0.001$). Detailed descriptions of the experiments are provided in the respective figure legends. All data presented in the figures are representative of at least three independent replicates and are expressed as means ± SD. Statistical analyses were performed in GraphPad Prism (GraphPad Software).

# Data availability

The RNA-Seq datasets were deposited in the BioProject accession PRJNA1225842. The datasets produced in this study are available in the following databases: RNA-Seq data: National Center for Biotechnology Information BioProject data (https://www.ncbi.nlm.nih.gov/sra/PRJNA1225842) and are publicly available as of the date of publication. The raw mass spectrometry data were irrevocably lost during a facility-

wide infrastructure upgrade and therefore cannot be deposited. However, the processed LC-MS/MS results, including peptide counts, Sequest HT scores, and related annotations, are provided as Dataset EV3. Sequence data from this article can be found in The *Arabidopsis* Information Resource (TAIR, http://www.arabidopsis.org) under the following accession numbers: *SRAS1.1* (AT5G66070; https://www.arabidopsis.org/locus?key=132297), *DSK2A* (AT2G17190; https://www.arabidopsis.org/locus?key=34510), *DSK2B* (AT2G17200; https://www.arabidopsis.org/locus?key=34511), *BES1* (AT3G50750; https://www.arabidopsis.org/locus?key=40370), *ATG8e* (AT2G45170; https://www.arabidopsis.org/locus?key=33868). The source data of this paper are collected in the following database record: biostudies:S-BSST1961 (https://www.ebi.ac.uk/biostudies/studies/S-BSST1961).

The source data of this paper are collected in the following database record: biostudies:S-SCDT-10_1038-S44319-025-00556-9.

## Peer review information

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

## Acknowledgements

We thank Beijing CapitalBio Technology Co., Ltd. for assistance with RNA-seq analysis. This work was supported by the Natural Science Foundation of Shandong Province (Grant Nos. ZR2024MC196 and ZR2022MC051), the Taishan Scholar Project of Shandong Province (Grant No. tsqn202408127), and the National Natural Science Foundation of China (Grant No. 32170306).

## Author contributions

**Xiao-Hu Li**: Data curation; Formal analysis; Validation; Investigation; Visualization; Methodology; Writing—original draft. **Meng Wang**: Data curation; Formal analysis; Validation; Investigation; Methodology. **Yi-Ran Xu**: Data curation; Formal analysis; Validation; Visualization; Methodology. **Qian-Huan Guo**: Validation; Visualization. **Peng Liu**: Methodology. **Chang-Ai Wu**: Project administration. **Guo-Dong Yang**: Project administration. **Jin-Guang Huang**: Resources; Project administration. **Shi-Zhong Zhang**: Resources; Supervision; Funding acquisition; Project administration. **Cheng-Chao Zheng**: Conceptualization; Resources; Supervision. **Kang Yan**: Conceptualization; Resources; Data curation; Formal analysis; Supervision; Writing—original draft; Project administration; Writing—review and editing.

Source data underlying figure panels in this paper may have individual authorship assigned. Where available, figure panel/source data authorship is listed in the following database record: biostudies:S-SCDT-10_1038-S44319-025-00556-9.

## Disclosure and competing interests statement

The authors declare no competing interests.

# Expanded View Figures

**Figure EV1.   The effects of SRAS1.1 on *Arabidopsis* drought tolerance.**

(**A–C**) Representative seedlings of wild-type, *SRAS1.1-14*, *sras1.1* mutants and *proSRAS1.1:SRAS1.1/sras1.1* complementation line (*COM1*) after 14 days of growth in 1/2 MS medium with or without 250 mM mannitol. Scale bars = 1 cm. Primary root length (**B**) and lateral root density (**C**) of seedlings shown in (**A**). Values represent means ± SD (*n* = 3 biological replicates), with 20 plants analyzed per replicate. (**B**) *P* values < 0.0001 (wild-type vs *SRAS1.1-14*), <0.0001 (wild-type vs *sras1.1*), 0.0217 (wild-type vs *COM1*), <0.0001 (*SRAS1.1-14* vs *sras1.1*), <0.0001 (*SRAS1.1-14* vs *COM1*) < 0.0001 (*sras1.1* vs *COM1*). (The following is the same order). (**C**) *P* values < 0.0001, < 0.0001, 0.8803, <0.0001, < 0.0001, 0.0013. (**D–I**) Phenotypic analysis of wild-type, *SRAS1.1-OE*, *sras1.1* mutants, and *COM1* seedlings grown on 1/2 MS medium with or without 250 mM mannitol (**D, G**). Images were taken 7 days after germination. Comparison of germination rates under normal conditions (**E, H**) and 250 mM mannitol treatment (**F, I**) between wild-type and transgenic plants. Values shown are means ± SD (*n* = 3 biological replicates). Significance was determined using Student's *t* test. (**E**) T = 24 h: *P* values > 0.9999 (wild-type vs *SRAS1.1-14*), >0.9999 (wild-type vs *SRAS1.1-26*), 0.9984 (wild-type vs *sras1.1*). (The following is the same order). T = 48 h: *P* values = 0.9999, 0.1836, >0.9999. T = 72 h: *P* values > 0.9999, >0.999, >0.9999. (**F**) T = 24 h: *P* values > 0.9999, >0.9999, >0.9999. T = 48 h: *P* values > 0.9999, >0.9999, >0.9999. T = 72 h: *P* values = 0.0092, 0.6742, 0.0476. T = 96 h: *P* values = 0.0067, 0.0083, 0.0347. T = 120 h: *P* values = 0.0044, 0.0057, 0.0433. T = 144 h: *P* values < 0.0001, <0.0001, 0.0191. T = 168 h: *P* values < 0.0001, <0.0001, 0.0207. (**H**) T = 24 h: *P* values = 0.6449 (wild-type vs *SRAS1.1-14*), 0.7372 (wild-type vs *sras1.1*), 0.0927 (wild-type vs *COM1*). (The following is the same order). T = 48 h: *P* values > 0.9999, 0.8942, 0.3219. T = 72 h: *P* values > 0.9999, >0.9999, >0.9999. (**I**) T = 24 h: *P* values > 0.9999, >0.9999, >0.9999. T = 48 h: *P* values > 0.9999, >0.9999, >0.9999. T = 72 h: *P* values = 0.0039, 0.0652, >0.9999. T = 96 h: *P* values = 0.0058, 0.0476, 0.7557. T = 120 h: *P* values = 0.0012, 0.0074, 0.9801. T = 144 h: *P* values = 0.0007, 0.0386, 0.9817. T = 168 h: *P* values < 0.0001, 0.0217, >0.9999. (**J–M**) Quantitative expression analysis of *DREB2A* (**J**), *RD20* (**K**), *RD29A* (**L**), and *RD26* (**M**). The data were normalized against *UBQ10* expression. Values shown are means ± SD (*n* = 3 biological replicates). (**J**) *P* values = 0.0212 (wild-type vs *SRAS1.1-14*), 0.0473 (wild-type vs *SRAS1.1-26*), <0.0001 (wild-type vs *sras1.1*), 0.9217 (*SRAS1.1-14* vs *SRAS1.1-26*) < 0.0001 (*SRAS1.1-14* vs *sras1.1*), <0.0001 (*SRAS1.1-26* vs *sras1.1*). (The following is the same order). (**K**) *P* values < 0.0001, <0.0001, <0.0001, 0.4539, <0.0001, <0.0001. (**L**) *P* values < 0.0001, <0.0001, <0.0001, 0.127, <0.0001, <0.0001. (**M**) *P* values = 0.0024, 0.0016, <0.0001, >0.9999, <0.0001, <0.0001. Data information: For (**B, C, J–M**) different lowercase letters represent significant differences, as determined by one-way ANOVA in combination with Tukey's multiple comparisons test (*P* < 0.05). For (**E, F, H, I**) significance was determined using Student's *t* test.

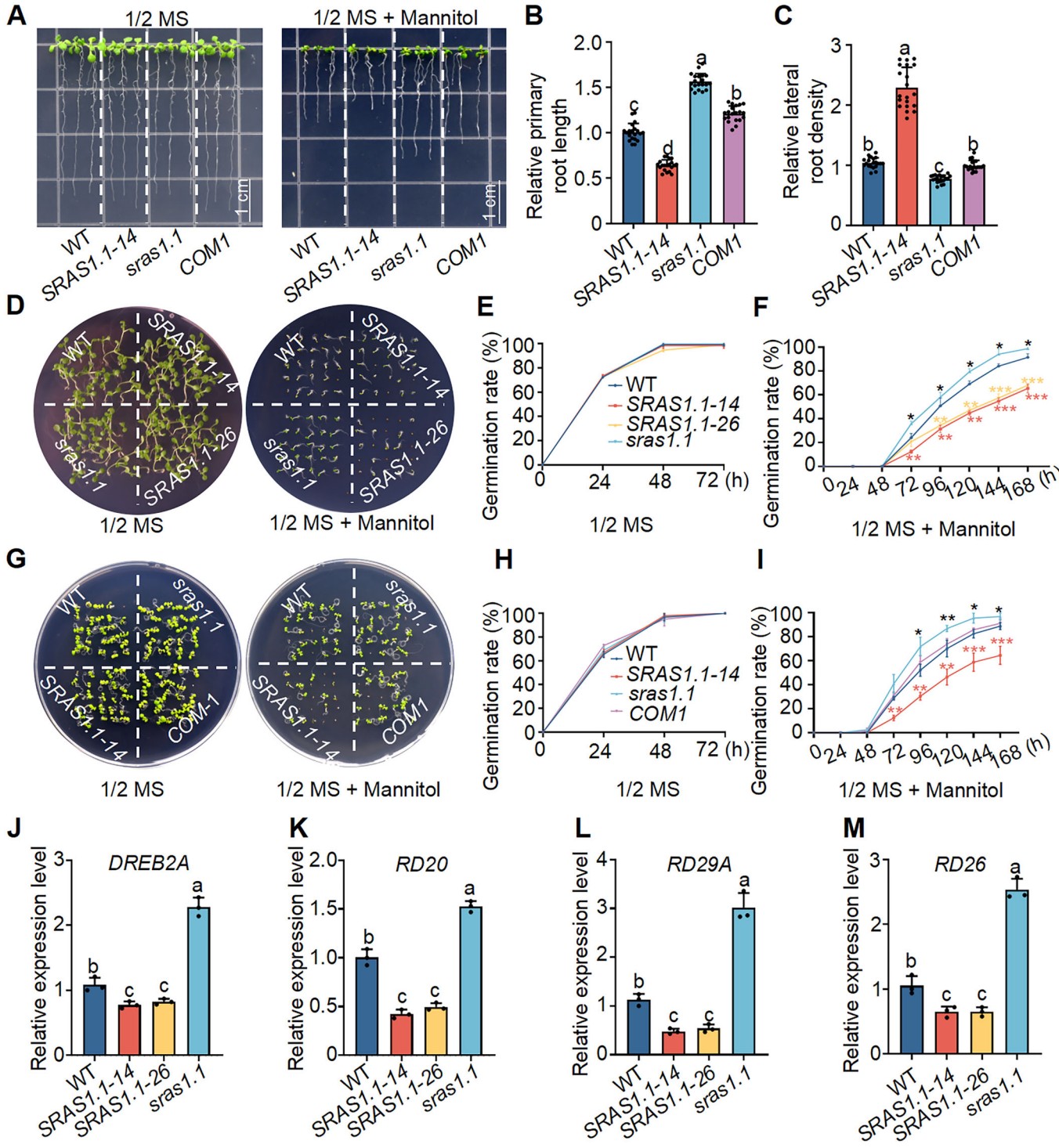

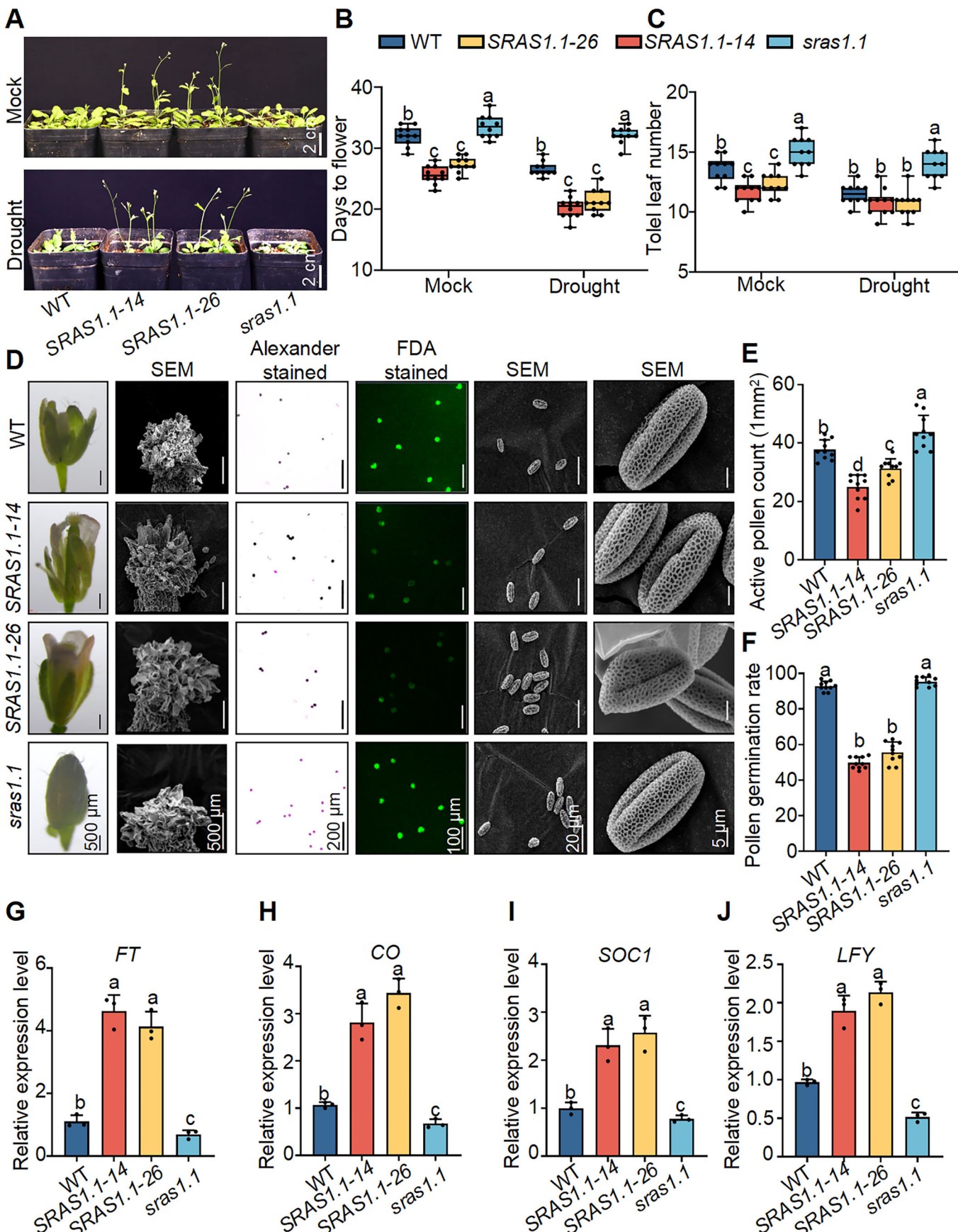

**Figure EV2.  Overexpression of *SRAS1.1* promotes flowering in *Arabidopsis*.**

(A) Morphology of wild-type and *SRAS1.1-14*, *SRAS1.1-26*, and *sras1.1* mutants at the flowering stage under control and drought conditions. Scale bars = 2 cm. (B, C) Total leaf number at flowering (B) and days to flowering (C) in wild-type, *SRAS1.1-14*, *SRAS1.1-26*, and *sras1.1* mutants grown with and without drought treatment. Data in (B, C) are plotted with box–whisker plots: the whiskers represent maximum and minimum values, and boxes represent the upper quartile, median, and lower quartile, dots represent data points. (B) Mock: *P* values < 0.0001 (wild-type vs *SRAS1.1-14*), <0.0001 (wild-type vs *SRAS1.1-26*), 0.0021 (wild-type vs *sras1.1*), 0.0917 (*SRAS1.1-14* vs *SRAS1.1-26*), <0.0001 (*SRAS1.1-14* vs *sras1.1*), <0.0001 (*SRAS1.1-26* vs *sras1.1*). (The following is the same order). Drought: *P* values < 0.0001, <0.0001, <0.0001, 0.4112, <0.0001, <0.0001. (C) Mock: *P* values = 0.0016, 0.0452, 0.0274, 0.5916, <0.0001, <0.0001. Drought: *P* values = 0.4801, 0.3633, <0.0001, 0.9968, <0.0001, < 0.0001. (D) Analysis of pollen grains from wild-type, *SRAS1.1-14*, *SRAS1.1-26*, and *sras1.1* mutants using alexander staining, fluorescein diacetate (FDA) staining, and scanning electron microscopy (SEM), respectively. (E, F) Number of FDA-stained pollen grains per 1 mm$^2$ (E), in vitro pollen germination rates (F) of wild-type, *SRAS1.1-14*, *SRAS1.1-26*, and *sras1.1* mutants. Values represent means ± SD (*n* = 3 biological replicates), with 10 plants analyzed per replicate. (E) *P* values < 0.0001 (wild-type vs *SRAS1.1-14*), 0.0104 (wild-type vs *SRAS1.1-26*), 0.0037 (wild-type vs *sras1.1*), 0.0297 (*SRAS1.1-14* vs *SRAS1.1-26*), <0.0001 (*SRAS1.1-14* vs *sras1.1*), <0.0001 (*SRAS1.1-26* vs *sras1.1*). (The following is the same order). (F) *P* values < 0.0001, <0.0001, 0.4716, 0.0576, <0.0001, <0.0001. (G–J) Quantitative expression analysis of *FT* (G), *CO* (H), *SOC1* (I), and *LFY* (J). The data were normalized against *UBQ10* expression. Values shown are means ± SD (*n* = 3 biological replicates). (G) *P* values < 0.0001 (wild-type vs *SRAS1.1-14*), <0.0001 (wild-type vs *SRAS1.1-26*), 0.0055 (wild-type vs *sras1.1*), 0.4145 (*SRAS1.1-14* vs *SRAS1.1-26*), <0.0001 (*SRAS1.1-14* vs *sras1.1*), <0.0001 (*SRAS1.1-26* vs *sras1.1*). (The following is the same order). (H) *P* values = 0.0002, <0.0001, 0.0313, 0.0711, <0.0001, <0.0001. (I) *P* values = 0.0011, 0.0003, 0.0389, 0.616, 0.0004, 0.0001. (J) *P* values < 0.0001, <0.0001, 0.0099, 0.1706, <0.0001, <0.0001. Data information: For (B, C, E, F, G–J) different lowercase letters represent significant differences, as determined by one-way ANOVA in combination with Tukey's multiple comparisons test (*P* < 0.05).

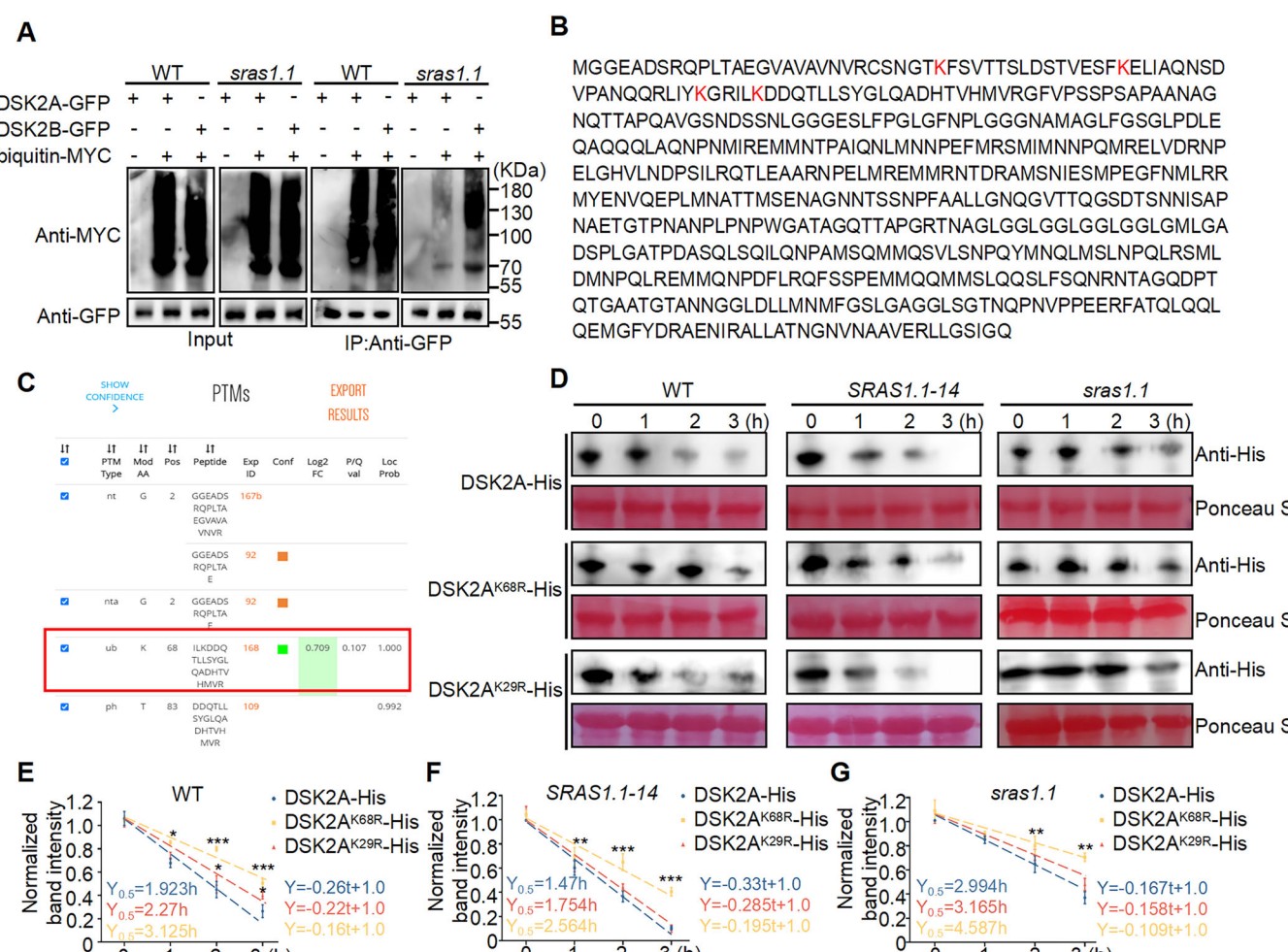

**Figure EV3. DSK2A is ubiquitinated by SRAS1.1, and Lys68 is the major ubiquitination site.**

(A) Ubiquitination of DSK2A-GFP and DSK2B-GFP in wild-type and *sras1.1* plants, detected in an *Arabidopsis* protoplast transient transformation assay. Anti-GFP antibody was used to immunoprecipitate DSK2A-GFP and DSK2B-GFP, and anti-MYC antibody was used to detect Ubiquitin-MYC. (B) Amino acid sequence of DSK2A with lysine residues highlighted in red. (C) Predicted post-translational modifications of DSK2A, with Lys68 indicated as the major ubiquitination site. (D) Degradation rates of DSK2A-His, DSK2AK68R-His, and DSK2AK29R-His in cell-free degradation assays using protein extracts from wild-type, *SRAS1.1-14*, and *sras1.1* mutant plants. Proteins were detected by immunoblotting with an anti-His antibody. Ponceau S staining was used as a loading control. (E–G) Quantified degradation rates of DSK2A-His (E), DSK2A$^{K68R}$-His (F), and DSK2A$^{K29R}$-His (G) plotted as linear regression curves. $Y_{0.5}$ denotes the time required for 50% degradation. Values shown are means ± SD ($n = 3$ biological replicates). All comparisons were made against DSK2A-His and analyzed by linear regression and Student's *t* test (*$P < 0.05$; **$P < 0.01$; ***$P < 0.001$). (E) T = 1 h: $P$ values = 0.0273 (DSK2A-His vs DSK2A$^{K68R}$-His), 0.3123 (DSK2A-His vs DSK2A$^{K29R}$-His). (The following is the same order). T = 2 h: $P$ values = 0.0009, 0.0433. T = 3 h: $P$ values = 0.0007, 0.0322. (F) T = 1 h: $P$ values = 0.0072, 0.4917. T = 2 h: $P$ values = 0.0005, 0.8971. T = 3 h: $P$ values < 0.0001, 0.3707. (G) T = 1 h: $P$ values = 0.2442, 0.9017. T = 2 h: $P$ values = 0.0074, 0.0981. T = 3 h: $P$ values = 0.0095, 0.0672.

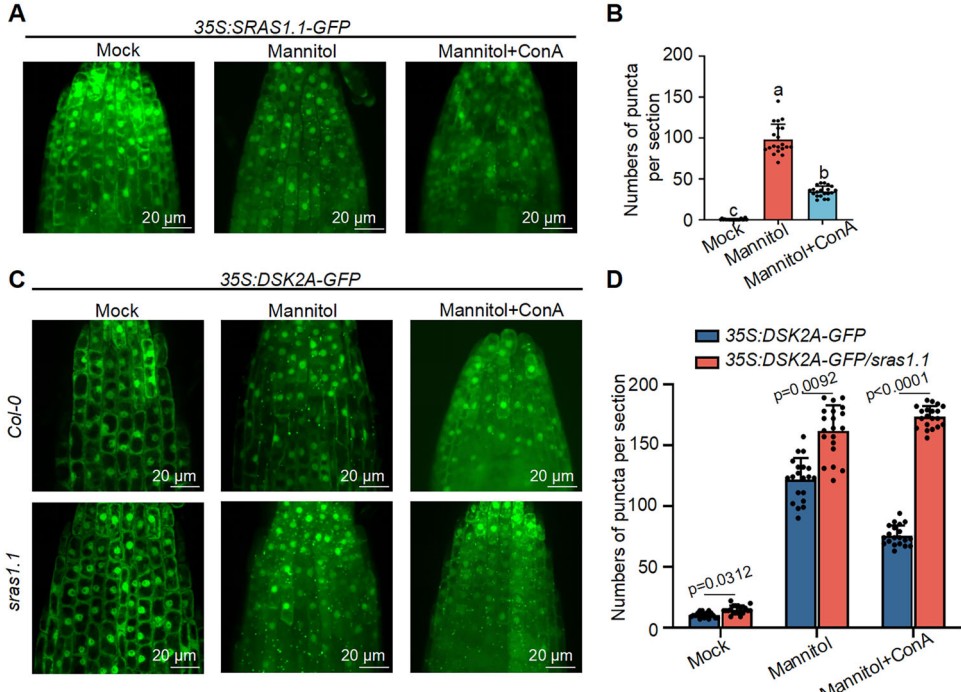

**Figure EV4. SRAS1.1 is involved in the regulation of cellular autophagy.**

(A) Confocal analysis of *35S:SRAS1.1-GFP* transgenic *Arabidopsis*. 5-day-old seedlings were exposed to 250 mM mannitol liquid medium for 30 min and then visualized by confocal laser scanning microscopy. Scale bars = 20 μm. (B) Numbers of puncta per section in the root cells of the *35S:SRAS1.1-GFP* transgenic *Arabidopsis* in (A). Values shown are means ± SD (*n* = 3 biological replicates), with 20 plants analyzed per replicate. *P* values < 0.0001 (Mock vs Mannitol), <0.0001 (Mock vs Mannitol + ConA), <0.0001 (Mannitol vs Mannitol + ConA). (C) Confocal analysis of *35S:DSK2A-GFP* and *35S:DSK2A-GFP/sras1.1* transgenic plants. Five-day-old seedlings were exposed to 250 mM mannitol liquid medium for 30 min and then visualized by confocal laser scanning microscopy. Scale bars = 20 μm. (D) Numbers of puncta per section in the root cells of the transgenic plants in (C). Values shown are means ± SD (*n* = 3 biological replicates), with 20 plants analyzed per replicate. Significance was determined using Student's *t* test. Mock: *P* values = 0.0312 (*35S:DSK2A-GFP* vs *35S:DSK2A-GFP/sras1.1*). Mannitol: *P* values = 0.0092. Mannitol + ConA: *P* values < 0.0001. Data information: For (B) different lowercase letters represent significant differences, as determined by one-way ANOVA in combination with Tukey's multiple comparisons test (*P* < 0.05). For (D) significance was determined using Student's *t* test.

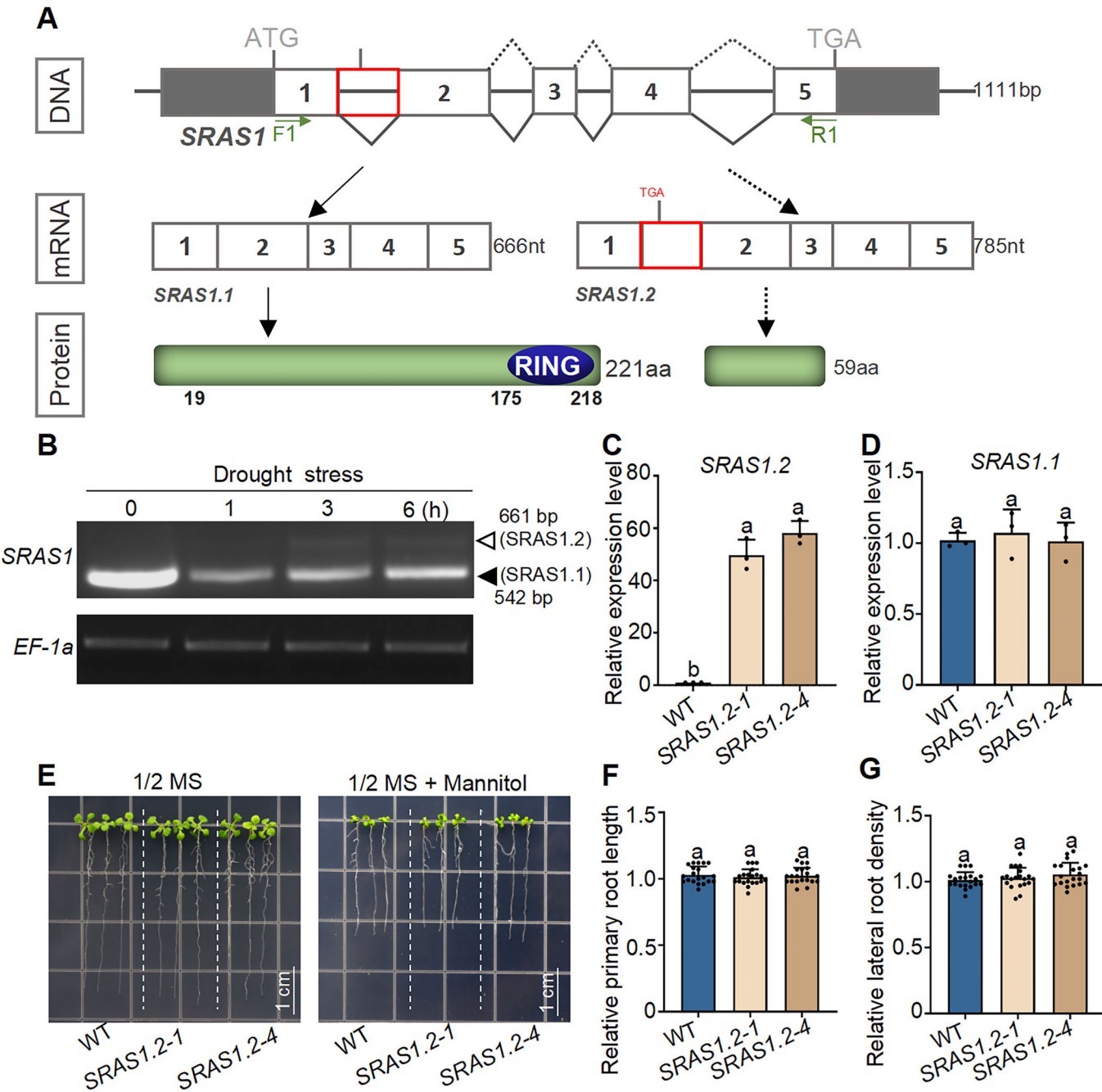

**Figure EV5.  Alternative splicing of *SRAS1* in response to drought stress.**

(A) Schematic diagram of *SRAS1* gene. Two different intron splice sites are indicated in the gene diagram. Arrows (F1 and R1) indicate the location of primers used in RT-PCR and green color represents the exon region; blue region represents the RING domain. (B) RT-PCR analysis of the expression levels of *SRAS1.1* and *SRAS1.2* at 0, 1, 3, and 6 h after 250 mM mannitol treatment. Elongation factor 1α (*EF-1α*) was used as an internal control. (C, D) Quantitative measurement of the expression levels of *SRAS1.2* (C) and *SRAS1.1* (D) in wild-type, *SRAS1.2* overexpressing plants (*SRAS1.2-1*, *SRAS1.2-4*). *UBQ10* was used as an internal control. Values shown are means ± SD (*n* = 3 biological replicates). (C) *P* values < 0.0001 (wild-type vs *SRAS1.2-1*), <0.0001 (wild-type vs *SRAS1.2-4*), 0.1163 (*SRAS1.2-1* vs *SRAS1.2-4*). (The following is the same order). (D) *P* values = 0.865, 0.9977, 0.8333. (E–G) Representative seedlings of wild-type, *SRAS1.2-1*, and *SRAS1.2-4* after 10 days of growth in 1/2 MS medium with or without 250 mM mannitol. Scale bars = 1 cm. Primary root length (F) and lateral root density (G) of seedlings shown in (E). Values shown are means ± SD (*n* = 3 biological replicates), with 20 plants analyzed per replicate. (F) *P* values = 0.6465 (wild-type vs *SRAS1.2-1*), 0.9411 (wild-type vs *SRAS1.2-4*), 0.8409 (*SRAS1.2-1* vs *SRAS1.2-4*). (The following is the same order). (G) *P* values = 0.7984, 0.1795, 0.4835. Data information: For (C, D, F, G) different lowercase letters represent significant differences, as determined by one-way ANOVA in combination with Tukey's multiple comparisons test (*P* < 0.05).

