## [Peer Review File · EMBO Reports]

SRAS1.1 E3 Ligase Mediates DSK2A Degradation to Regulate Autophagy and Drought Tolerance in Arabidopsis

Xiaohu Li, Meng Wang, Yiran Xu, Qianhuan Guo, Peng Liu, Changai Wu, Guodong Yang, Jinguang Huang, Shizhong Zhang, ChengChao Zheng, and Kang Yan

Corresponding author(s): Kang Yan (kangyan@sdau.edu.cn), ChengChao Zheng (cczheng@sdau.edu.cn), Shizhong Zhang (shizhong@sdau.edu.cn)

Review Timeline:

Submission Date:	24th Dec 24
Editorial Decision:	14th Feb 25
Revision Received:	29th Apr 25
Editorial Decision:	2nd Jun 25
Revision Received:	16th Jun 25
Accepted:	21st Jul 25

Transaction Report:

Dear Prof. Yan

Thank you for the submission of your research manuscript to our journal. I sincerely apologize for the delay in handling your manuscript, but we have now received the two enclosed reports.

As you will see, the referees acknowledge that the findings are interesting and that the conclusions are overall supported by the data presented but they also raise a number of concerns and have suggestions how to further strengthen the data that need to be addressed.

Given the constructive and supportive comments, we would like to invite you to revise your manuscript with the understanding that the referee concerns (as detailed above and in their reports) must be fully addressed and their suggestions taken on board. Please address all referee concerns in a complete point-by-point response. Acceptance of the manuscript will depend on a positive outcome of a second round of review. It is EMBO Reports policy to allow a single round of revision only and acceptance or rejection of the manuscript will therefore depend on the completeness of your responses included in the next, final version of the manuscript.

We realize that it is difficult to revise to a specific deadline. In the interest of protecting the conceptual advance provided by the work, we recommend a revision within 3 months (May 14). Please discuss the revision progress ahead of this time with the editor if you require more time to complete the revisions.

I am also happy to discuss the revision further via e-mail or a video call, if you wish.

*****IMPORTANT NOTE:

We perform an initial quality control of all revised manuscripts before re-review. Your manuscript will FAIL this control and the handling will be delayed IN CASE the following APPLIES:

- 1) A data availability section providing access to data deposited in public databases is missing. If you have not deposited any data, please add a sentence to the data availability section that explains that.
- 2) Your manuscript contains statistics and error bars based on $n=2$. Please use scatter blots in these cases. No statistics should be calculated if $n=2$.

When submitting your revised manuscript, please carefully review the instructions that follow below. Failure to include requested items will delay the evaluation of your revision.*****

- 1) a .docx formatted version of the manuscript text (including legends for main figures, EV figures and tables). Please make sure that the changes are highlighted to be clearly visible.
- 2) individual production quality figure files as .eps, .tif, .jpg (one file per figure). Please download our Figure Preparation Guidelines (figure preparation pdf) from our Author Guidelines pages <https://www.embopress.org/page/journal/14693178/authorguide> for more info on how to prepare your figures.
- 3) a .docx formatted letter INCLUDING the reviewers' reports and your detailed point-by-point responses to their comments. As part of the EMBO Press transparent editorial process, the point-by-point response is part of the Review Process File (RPF), which will be published alongside your paper.
- 4) a complete author checklist, which you can download from our author guidelines (). Please insert information in the checklist that is also reflected in the manuscript. The completed author checklist will also be part of the RPF.
- 5) Please note that all corresponding authors are required to supply an ORCID ID for their name upon submission of a revised manuscript (). Please find instructions on how to link your ORCID ID to your account in our manuscript tracking system in our Author guidelines

()

6) We replaced Supplementary Information with Expanded View (EV) Figures and Tables that are collapsible/expandable online. A maximum of 5 EV Figures can be typeset. EV Figures should be cited as 'Figure EV1, Figure EV2' etc... in the text and their respective legends should be included in the main text after the legends of regular figures.

7) Before submitting your revision, primary datasets (and computer code, where appropriate) produced in this study need to be deposited in an appropriate public database (see < <https://www.embopress.org/page/journal/14693178/authorguide#dataavailability>>). Specifically, we would kindly ask you to provide public access to the RNA-seq datasets.

The accession numbers and database should be listed in a formal "Data Availability " section (placed after Materials & Method) that follows the model below (see also < <https://www.embopress.org/page/journal/14693178/authorguide#dataavailability>>). Please note that the Data Availability Section is restricted to new primary data that are part of this study.

Data availability

Additional information on source data and instruction on how to label the files are available .

10) Figure legends and data quantification:

- the name of the statistical test used to generate error bars and P values,
- the number (n) of independent experiments (please specify technical or biological replicates) underlying each data point,
- the nature of the bars and error bars (s.d., s.e.m.)

- If the data are obtained from n {less than or equal to} 5, show the individual data points in addition to the SD or SEM.
- If the data are obtained from n {less than or equal to} 2, use scatter blots showing the individual data points.

11) Our journal encourages inclusion of *data citations in the reference list* to directly cite datasets that were re-used and obtained from public databases. Data citations in the article text are distinct from normal bibliographical citations and should directly link to the database records from which the data can be accessed. In the main text, data citations are formatted as follows: "Data ref: Smith et al, 2001" or "Data ref: NCBI Sequence Read Archive PRJNA342805, 2017". In the Reference list, data citations must be labeled with "[DATASET]". A data reference must provide the database name, accession number/identifiers and a resolvable link to the landing page from which the data can be accessed at the end of the reference. Further instructions are available at .

12) All Materials and Methods need to be described in the main text using our 'Structured Methods' format. According to this format, the Methods section includes a Reagents and Tools Table (listing key reagents, experimental models, software and relevant equipment and including their sources and relevant identifiers) followed by a Methods and Protocols section describing the methods, ideally using a step-by-step protocol format. The aim is to facilitate adoption of the methodologies across labs. Please download and fill our Reagents and Tools Table template (.docx), which you can find in our author guidelines: <https://www.embopress.org/page/journal/14693178/authorguide#structuredmethods>. When submitting your revised manuscript, please do not include the Reagents and Tools Table in the Methods section of the manuscript but upload it as a separate file choosing the file type "Reagent Table". An example of a Method paper with Structured Methods can be found here: <https://www.embopress.org/doi/10.15252/msb.20178071>.

13) As part of the EMBO publication's Transparent Editorial Process, EMBO Reports publishes online a Review Process File to accompany accepted manuscripts. This File will be published in conjunction with your paper and will include the referee reports, your point-by-point response and all pertinent correspondence relating to the manuscript.

Yours sincerely,

=====

Referee #1:

The manuscript by Li et al. presents an important and innovative study on the role of E3 ubiquitin ligases in regulating autophagy under drought stress in Arabidopsis. The study provides valuable insights into the interplay between the ubiquitin-proteasome system (UPS) and autophagy, both of which are crucial for cellular homeostasis and stress response mechanisms in plants. The authors have performed a series of well-executed experiments that demonstrate SRAS1.1's role in modulating autophagy and regulating drought tolerance through the degradation of the autophagy receptor DSK2A. The molecular data, including transcriptome analysis and biochemical assays, are solid, and the visual data from subcellular localization experiments offer further credibility to the study's conclusions.

Overall, the manuscript is well-written and scientifically rigorous, presenting compelling evidence that SRAS1.1 negatively regulates drought stress tolerance by targeting DSK2A. I believe the manuscript is a strong candidate for publication in EMBO Reports with minor revisions.

Specific Comments:

1) Introduction Section:

The Introduction could benefit from further background on the biological function of SRAS1.1. While the authors describe the gene and its connection to E3 ligase activity, it would be helpful to include more detail on its general role in plant stress responses. Providing context on its evolutionary conservation or its functional implications in other organisms might also enrich the introduction.

2) Discussion Section:

The Discussion would benefit from subtitles to enhance clarity and guide the reader through the various aspects of the findings. This would help in distinguishing the sections focusing on autophagy regulation, drought tolerance, and brassinosteroid signaling, making the discussion more accessible.

3) The mention of brassinosteroid signaling is intriguing, but the discussion could be expanded to explain its potential interaction with autophagy in drought tolerance more comprehensively. A deeper exploration of this connection could shed light on how these signaling pathways overlap and contribute to stress resilience in plants.

4) While the study focuses on Arabidopsis, it would be helpful for the authors to briefly discuss whether SRAS1.1 or its homologs are present in other crops, particularly those of agricultural importance. Highlighting how these findings might be translated to improve drought tolerance in crops could broaden the impact and relevance of the research. A forward-looking statement in the Discussion could add an important dimension to the paper's conclusion.

5) The authors mention a link between early flowering and drought tolerance. It would be helpful to clarify whether they have examined changes in flowering-related gene expression in SRAS1.1 plants to confirm that the gene does indeed regulate early flowering. This additional data would strengthen the claim and provide a more direct connection between flowering time and drought tolerance in the context of SRAS1.1 activity.

6) In this study, authors observed that SRAS1.1 relocates from the nucleus to the cytoplasm in response to drought stress, which further associates with autophagosomes, and modulates autophagy-related gene expression and BES1 accumulation. This point is a very important to address the molecular mechanism of SRAS1.1 in plant drought tolerance. Therefore, only the transient expression experiments using tobacco is not enough. Using 35S: SRAS1.1-GFP stable transgenic line for laser confocal microscopy analysis or conducting western blot analysis combined with nuclear cytoplasmic separation should be very helpful.

7) Figure Clarifications:

a) Figure 2B: The meaning of the numerical values in this figure should be clarified. It is not entirely clear what the values represent, and a more detailed legend would help readers interpret the data.

b) Figures 2D and 2F: The KD and KDa values appear inconsistent across these figures. For clarity and consistency, I suggest standardizing the units and ensuring that the values are properly aligned in both the figure panels and the figure legend.

8) Minor Editorial Issues:

a) Line 95: The full name of BES1 should be spelled out as "BRI1-EMS-SUPPRESSOR 1" for the first mention.

b) Lines 137-138: The description "longer" is not suitable in this context. Under normal conditions, the flowering time of SRAS1.1-OE plants was 4.26 days shorter than that of wild-type plants. The wording should reflect this difference in timing more clearly.

Referee #2:

The authors elucidated the mechanism by which the E3 ligase SRAS1.1 mediates the degradation of DSK2A, thereby regulating autophagy and drought tolerance in Arabidopsis. This study establishes a connection between ubiquitination and autophagy in plant responses to drought, which may be of interest to readers within this field. However, several issues need to be addressed.

1) All phenotypes were characterized using a single mutant allele, raising concerns about the potential for T-DNA off-target effects. Data from complementary lines should be included in each experiment to validate its role in drought tolerance.

2) In the key data presented in Fig. 1D, the root lengths of WT and *sras1.1* appear similar, which contradicts the quantitative data shown in Fig. 1F.

3) In Figs. 1J-F, RNA-Seq data are only provided for the WT and *sras1.1* mutant; but the data from overexpressing plants are absent. Comparisons among these three samples need to be clearly illustrated.

4) In Figs. 2B and 2E, it is advisable to include another protein with similar localization as part of the control groups rather than solely relying on empty vectors as negative controls.

5) The ubiquitination data presented in Fig. EV5A would benefit from being relocated to the main figures, with all immunoblots to demonstrate components within this system. Furthermore, an *in vivo* ubiquitination assay should also be conducted to substantiate these conclusions.

6) In Fig. 4A, while localization of SRAS1.1 is demonstrated, it would be crucial to show DSK2A's localization under the *sras1.1* background with these conditions.

7) Given that SRAS1 undergoes alternative splicing, it is essential to characterize how drought influences its splicing patterns. Characterization of both *sras1* mutants and overexpressing lines should also be included to confirm the specific isoform used.

8) The figure legends currently lack detail; providing additional information could enhance understanding significantly.

Responses to Referee #1

Thank you very much for your comments. We have conducted all the additional experiments requested, including subcellular localization of *35S:SRAS1.1-GFP*, expression analysis of flowering-related genes, phenotypic analyses, and *in vivo* ubiquitination assays. In addition, we have further revised the manuscript according to your suggestions. We believe that your constructive suggestions have substantially enhanced the rigor and clarity of our study.

1) The Introduction could benefit from further background on the biological function of SRAS1.1. While the authors describe the gene and its connection to E3 ligase activity, it would be helpful to include more detail on its general role in plant stress responses. Providing context on its evolutionary conservation or its functional implications in other organisms might also enrich the introduction.

Thank you for the suggestion. We have added additional background information on the biological function of *SRAS1.1* in the Introduction (line 107–110). Furthermore, to address the reviewer's comments regarding its evolutionary conservation and potential functional implications in other organisms, we have included a discussion of these aspects in the Discussion section (line 404–411).

2) The Discussion would benefit from subtitles to enhance clarity and guide the reader through the various aspects of the findings. This would help in distinguishing the sections focusing on autophagy regulation, drought tolerance, and brassinosteroid signaling, making the discussion more accessible.

Thank you for your comments. We fully agree that the use of subtitles would improve clarity and help guide the reader through different aspects of the discussion. However, in accordance with the formatting requirements, we have not included explicit subheadings. Instead, we have revised the structure of the Discussion section by separating it into clearly defined paragraphs and introducing topic sentences at the beginning of each to improve readability and highlight the main themes of our findings.

3) The mention of brassinosteroid signaling is intriguing, but the discussion could be expanded to explain its potential interaction with autophagy in drought tolerance more comprehensively. A deeper exploration of this connection could shed light on how these signaling pathways overlap and contribute to stress resilience in plants.

This is a good point. In the revised manuscript, we have expanded our discussion on the potential interplay between brassinosteroid signaling and autophagy in the context of drought stress (line 390–404). We highlighted recent studies showing that BR signaling components such as BIN2 and BES1 interact with autophagy pathways, and we discussed how BR-related gene expression and phenotypes observed in the *sras1.1* mutant support a functional connection between these pathways. We hope this

expanded discussion sufficiently addresses your comment.

4) While the study focuses on Arabidopsis, it would be helpful for the authors to briefly discuss whether SRAS1.1 or its homologs are present in other crops, particularly those of agricultural importance. Highlighting how these findings might be translated to improve drought tolerance in crops could broaden the impact and relevance of the research. A forward-looking statement in the Discussion could add an important dimension to the paper's conclusion.

Thank you for your suggestion. In the revised manuscript, we have included a forward-looking statement at the end of the Discussion to highlight the broader relevance of our findings (line 412–416). Specifically, we noted that SRAS1 belongs to the ATL (*Arabidopsis* Tóxicos en Levadura) subfamily of RING-type E3 ubiquitin ligases, which is highly conserved across diverse plant species. We also emphasized the potential of leveraging SRAS1.1 and its homologs through genetic engineering to enhance crop drought resilience (line 405–411). These additions underscore the translational value of our study and address the reviewer's concern regarding its applicability to agriculturally important species.

5) The authors mention a link between early flowering and drought tolerance. It would be helpful to clarify whether they have examined changes in flowering-related gene expression in SRAS1.1 plants to confirm that the gene does indeed regulate early flowering. This additional data would strengthen the claim and provide a more direct connection between flowering time and drought tolerance in the context of SRAS1.1 activity.

According to your suggestion, we examined the expression levels of key flowering-related genes, including *FLOWERING LOCUS T (FT)*, *CONSTANS (CO)*, *SUPPRESSOR OF OVEREXPRESSION OF CO 1 (SOC1)*, and *LEAFY (LFY)* in *SRAS1.1* overexpressing plants. The results showed expression patterns consistent with the observed early flowering phenotype. These data further support the role of *SRAS1.1* in regulating flowering time under drought stress. The detailed results have been added to Figure EV2G–J in the revised manuscript (line 157–160).

6) In this study, authors observed that SRAS1.1 relocates from the nucleus to the cytoplasm in response to drought stress, which further associates with autophagosomes, and modulates autophagy-related gene expression and BES1 accumulation. This point is a very important to address the molecular mechanism of SRAS1.1 in plant drought tolerance. Therefore, only the transient expression experiments using tobacco is not enough. Using 35S: SRAS1.1-GFP stable transgenic line for laser confocal microscopy analysis or conducting western blot analysis combined with nuclear cytoplasmic separation should be very helpful.

Thank you for this valuable suggestion. Following your recommendation, we examined the subcellular localization of SRAS1.1 using *35S:SRAS1.1-GFP* stable transgenic *Arabidopsis* lines under different durations of drought stress. Consistent with the transient expression results in *N. benthamiana*, *35S:SRAS1.1-GFP* accumulated in the cytoplasm and formed distinct punctate structures under drought conditions. Upon treatment with the autophagy inhibitor concanamycin A (ConA), *35S:SRAS1.1-GFP* progressively relocalized to the nucleus. The detailed results have been added to (Fig. EV4A,B) in the revised manuscript (line 301–303).

7) Figure Clarifications:

a) Figure 2B: The meaning of the numerical values in this figure should be clarified. It is not entirely clear what the values represent, and a more detailed legend would help readers interpret the data.

b) Figures 2D and 2F: The KD and kDa values appear inconsistent across these figures. For clarity and consistency, I suggest standardizing the units and ensuring that the values are properly aligned in both the figure panels and the figure legend.

Thank you for your suggestion. We have revised the figure legend of Figure 2B to clarify the meaning of the numerical values and ensure the data are clearly interpretable. According to your suggestion, we have standardized the use of “kDa” across all figure panels and legends for clarity and consistency.

8) Minor Editorial Issues:

a) Line 95: The full name of BES1 should be spelled out as "BRI1-EMS-SUPPRESSOR 1" for the first mention.

b) Lines 137-138: The description "longer" is not suitable in this context. Under normal conditions, the flowering time of SRAS1.1-OE plants was 4.26 days shorter than that of wild-type plants. The wording should reflect this difference in timing more clearly.

Thank you very much for your suggestions. We have corrected the full name of BES1 to "BRI1-EMS-SUPPRESSOR 1" at its first mention (line 105), and revised the wording in line 151–152 to accurately reflect the earlier flowering time of *SRAS1.1-OE* plants.

Responses to Referee #2:

Thank you very much for your helpful comments. We have performed all the additional experiments as requested, including the *in vivo* ubiquitination assay and stress-related phenotypic analyses. Furthermore, we have thoroughly revised the manuscript according to your suggestions. We believe that your constructive suggestions have significantly helped us improve and further develop the scientific narrative of our study.

1) All phenotypes were characterized using a single mutant allele, raising concerns about the potential for T-DNA off-target effects. Data from complementary lines should be included in each experiment to validate its role in drought tolerance.

Thank you for your comment. We have addressed this concern by analyzing *proSRAS1.1:SRAS1.1/sras1.1* complementation lines (*COM1*). The complemented plants restored drought-sensitive phenotypes, including reduced germination, shorter roots, and lower survival rates, similar to the wild-type. These data, consistent with wild-type *SRAS1.1* expression, confirm the specificity of the observed phenotypes. The detailed results have been added to Figure EV1 and Appendix Figure S1 (line 138–141).

2) In the key data presented in Fig. 1D, the root lengths of WT and sras1.1 appear similar, which contradicts the quantitative data shown in Fig. 1F.

Thank you for pointing this out. Upon careful re-examination, we found that the dashed lines in Figure 1D were incorrectly marked, leading to a potential misinterpretation of the root length comparison. We have corrected the labeling in the revised figure to ensure consistency and clarity. This phenotype was validated in the complementation lines (Fig. EV1A–C), further confirming the role of *SRAS1.1* in regulating root growth under drought conditions.

3) In Figs. 1J-F, RNA-Seq data are only provided for the WT and sras1.1 mutant; but the data from overexpressing plants are absent. Comparisons among these three samples need to be clearly illustrated.

Thank you for your suggestion. In response, we have re-analyzed the RNA-seq data and added results from the *SRAS1.1* overexpression lines alongside those of the wild-type and *sras1.1* mutant. These data are now integrated into the revised Figure 2 to enable a direct and comprehensive comparison among the three genotypes (line 169–178).

4) In Figs. 2B and 2E, it is advisable to include another protein with similar localization as part of the control groups rather than solely relying on empty vectors as negative controls.

Thank you for your comments and suggestions. In the revised Figures 3B and 3E (originally Figures 2B and 2E), we have included CSN5A as a positive control and ABI5 as a negative control. These additions help confirm the specificity of SRAS1.1 interactions with DSK2A and DSK2B. The results are shown in Figure 3E and Appendix Figure S3.

5) *The ubiquitination data presented in Fig. EV5A would benefit from being relocated to the main figures, with all immunoblots to demonstrate components within this system. Furthermore, an in vivo ubiquitination assay should also be conducted to substantiate these conclusions.*

This is a good point and is critical for supporting the conclusion that SRAS1.1 mediates the ubiquitination and degradation of DSK2A. According to your suggestion, we have moved the *in vitro* ubiquitination assay (original Fig. EV5A) into the main figures (now Fig. 4B). Additionally, we performed an *in vivo* ubiquitination assay to further validate the role of SRAS1.1 in regulating DSK2A stability (Fig. EV3A).

Furthermore, we generated a monoclonal antibody against DSK2 and used it to detect endogenous DSK2A protein levels in wild-type, *SRAS1.1* overexpression, and *sras1.1* mutant plants. As shown in the revised Figure 4A, DSK2A levels were markedly reduced in overexpression lines and increased in *sras1.1* mutants compared to wild-type, further supporting our conclusion. The detailed results have been added to (Figs. 4A,B and EV3A) in the revised manuscript (line 207–222).

6) *In Fig. 4A, while localization of SRAS1.1 is demonstrated, it would be crucial to show DSK2A's localization under the sras1.1 background with these conditions.*

Thanks for your suggestions. In accordance with your comment, we examined the subcellular localization of *35S:DSK2A-GFP* in both wild-type and *sras1.1* backgrounds, with or without 250 mM mannitol treatment. Under drought stress, *35S:DSK2A-GFP* formed punctate structures consistent with *35S:SRAS1.1-GFP*. Notably, fluorescence intensity of *35S:DSK2A-GFP* was markedly increased in *sras1.1* mutants. These results have been incorporated into the revised manuscript as Figure EV4C,D (line 353–355).

7) *Given that SRAS1 undergoes alternative splicing, it is essential to characterize how drought influences its splicing patterns. Characterization of both sras1 mutants and overexpressing lines should also be included to confirm the specific isoform used.*

Thank you for your suggestion. We analyzed how drought stress influences the splicing pattern of *SRAS1* (Fig. EV5A,B). The results showed that the expression level of *SRAS1.2* remained low under drought conditions. To further clarify the functional relevance of the isoforms, we generated overexpression lines for *SRAS1.2* and assessed their phenotypes under drought stress (Fig. EV5C–G; Appendix Figure

S9). Compared to wild-type plants, *SRAS1.2-OE* lines exhibited no significant differences in germination rate, primary root length, or survival rate. These findings indicate that *SRAS1.2* does not play a major functional role under drought stress conditions. These results have been incorporated into the revised manuscript as Figure EV5 and Appendix Figure S9 (line 373–376).

Importantly, these results also help to clarify the functional divergence between *SRAS1.1* and *SRAS1.2*, providing further support for the model that *SRAS1.1* is the primary isoform mediating drought-induced autophagic responses. This clarification strengthens our interpretation of the distinct roles of DSK2A and DSK2B in SRAS1-mediated regulation (line 376–385).

8) The figure legends currently lack detail; providing additional information could enhance understanding significantly.

Thank you for pointing this out. We have carefully revised all figure legends in the manuscript to provide more comprehensive descriptions of the data, annotations, and experimental details. The updated captions now ensure clarity, completeness, and improved readability.

Dear Prof. Yan,

Thank you for the submission of your revised manuscript to EMBO reports. It was evaluated by former referee #2 who considers all concerns adequately addressed and supports publication.

Browsing through the manuscript myself, I noticed a few editorial things that we need before we can proceed with the official acceptance of your study.

- Please remove the Data availability statement on the title page, line 25-26.
- It is not required to list the ORCID IDs on the title page and these can be removed as well.
- Regarding the Author Contributions, we now use CRediT to specify the contributions of each author in the journal submission system. Therefore, please remove the Author Contributions from the manuscript file and make sure that the author contributions in our online manuscript tracking system are correct and up-to-date. The information you specified in the system will be automatically retrieved and typeset into the article. You can enter additional information in the free text box provided, if you wish.
- We need the ORCID IDs for all corresponding authors, but these are currently missing for Dr. ChengChao Zheng and Dr. Shizong Zhang. Please find instructions on how to link your ORCID ID to your account in our manuscript tracking system in our Author guidelines (<https://www.embopress.org/page/journal/14693178/authorguide#authorshipguidelines>)
- Please upload the main and EV figures as individual production quality Figure files; the individual Figure files should not have legends and main and EV figure legends should only be in the manuscript not on the figure.
- Appendix Tables S1-S3 should be uploaded as individual .xls files and called Dataset EV1-EV3. Please provide a legend in a separate tab and please do not forget to change the callouts in the text.
- Appendix Table S4 should be Table EV1 and again, the legend needs to be part of the file.
- Appendix: while the table of content lists page numbers, these are missing on the subsequent pages.
- Please remove the Appendix figure legends from the main manuscript file. We only need them in the Appendix.
- Appendix figure S1A: please define the scale bar size only in the legend and remove the text (2 cm) from the scale bar itself.
- Appendix figure S2A: an arrow is defined in the legend but seems absent from the panel.
- Please provide a scale bar for Appendix Figure S3A.
- Appendix Figure S4: you define the statistical test for all panels and describe significant difference with lowercase letters but it appears that the description of "*" for S4F is missing.
- Appendix figure S4H: please define the boxplot (max, min, line, whiskers). Currently it is defined as mean plus/minus SD, which appears inaccurate.
- The scale bar in Appendix Figure S5A is difficult to see. Please change the color or line width and please define the size only in the legend, not on the scale bar itself.
- Is "means plus/minus SD" appropriate for the boxplots shown in Appendix Figure S5B?
- Appendix Figure S8A and S9D: the scale bars are difficult to see and please define the size only in the legend not on the scale bar itself. S8B: please provide a scale bar.
- The manuscript sections should be in the following order: Title page - Abstract & Keywords - Introduction - Results - Discussion - Methods - Data Availability - Acknowledgments - Disclosure Statement & Competing Interests - References - Figure Legends - (Main Tables with legends if applicable) - Expanded View Figure Legends.
- Please remove the instructions text from the Reagents and Tools table ("Instructions: Please complete the relevant fields below, adding rows as needed. The following page provides an example of a completed table and additional instruction for entering your data in the table.")
- Author Checklist: does Line 57, experimental animals and model organisms indeed apply?
- Author Checklist, line 82. Please include a statement on blinding in the methods section, as indicated.
- Please deposit the mass spectrometry data in a public repository and provide the accession number and URL in the Data availability section.
- Our production/data editors have asked you to clarify several points in the figure legends (see below). Please incorporate these changes in the manuscript and return the revised file with tracked changes with your final manuscript submission.

A) Statistical test information. Only p-values that are actually shown in the figure panel(s) should (and must) be defined in the legends, all others should be removed from (or added to) the legend. Moreover, we ask for the specification of exact p-values:
- Please note that the exact p values are not provided in the legends of figures 1A, C, E, F, H, I, K, L; 2F, G, H, I; 3C, 4E, F; 5B, C, E, F, H, I; 6C, D, E; EV1 B, C, F, I, J-M; EV2 B, C, E, F, G-J; EV3 E-G; EV4 B, D; EV5 C, D, F, G
- Please indicate the statistical test used for data analysis in the legends of figures 2A, B

B) Replicates and error bars:

- Please note that the box plots need to be defined in terms of minima, maxima, centre, bounds of box and whiskers, and percentile in the legends of figures 1E, F, K; 5B, C.
- Please note that information related to n is missing in the legends of figures 2A, B

- The SRAS1.1+ABI5.czi source data file seems to be corrupted? In contrast to the other .czi files you provided for Figure 3E, I could not open it in Fiji. Please check.

Just to make sure, the YFP channel for cYFP+DSK2A-nYFP, cYFP+DSK2B-nYFP, and SRAS1.1+ABI5 appears all black, also in the source data. Just in case there was any background offset applied, please specify so.

In line with this and for appropriate documentation, could you please provide the source data for Appendix Figure S8? Thank you very much.

- Please provide specific URLs for AT5G66070, AT2G17190, AT2G17200, AT3G50750, AT2G45170, S-BSST196 in the data availability statement.

- Please upload the source data as one .zip folder per figure.

- As a standard procedure we modify the abstract to make it more accessible to our general readership. Please find my suggestion below my signature.

With kind regards,

Martina Rembold

=====

Referee #2:

The authors have conducted additional experiments to address my inquiries.

=====

Drought stress significantly impacts plant growth and productivity, requiring complex adaptive responses to ensure survival. In eukaryotes, autophagy and the ubiquitin-proteasome system (UPS) are critical pathways for maintaining cellular homeostasis under stress. While their interaction is well-studied in animals, it remains poorly understood in plants, particularly under drought conditions. Here, we identify the E3 ubiquitin ligase SRAS1.1 as a negative regulator of drought tolerance in Arabidopsis, mediating its function through the ubiquitination and degradation of the autophagy receptor DSK2A. Loss of SRAS1.1 enhances drought tolerance by reducing water loss, increasing survival rates, and accelerating flowering. SRAS1.1 directly interacts with and ubiquitinates the autophagy receptor DSK2A, promoting its degradation via the 26S proteasome. Under drought stress, SRAS1.1 relocates from the nucleus to the cytoplasm, associates with autophagosomes, induces the accumulation of BES1, a regulator of the brassinosteroid pathway, and modulates autophagy-related gene expression. These findings provide novel insights into UPS-autophagy crosstalk in plants and highlight SRAS1.1 as a promising target for genetic engineering to develop drought-resilient crops and to advance sustainable agriculture.

All editorial and formatting issues were resolved by the authors.

Prof. Kang Yan
Shandong Agricultural University
College of Life Sciences
Daizong St 61
Tai'an, Shandong 271018
China

Dear Prof. Yan,

I am very pleased to accept your manuscript for publication in the next available issue of EMBO reports. Thank you for your contribution to our journal.

Yours sincerely,
